# Midbrain dopamine neurons signal aversion in a reward-context-dependent manner

**Hideyuki Matsumoto[1,2], Ju Tian[1,2], Naoshige Uchida[1,2], Mitsuko Watabe-Uchida[1,2]\***

[1]Center for Brain Science, Harvard University, Cambridge, United States; [2]Department of Molecular and Cellular Biology, Harvard University, Cambridge, United States

**Abstract** Dopamine is thought to regulate learning from appetitive and aversive events. Here we examined how optogenetically-identified dopamine neurons in the lateral ventral tegmental area of mice respond to aversive events in different conditions. In low reward contexts, most dopamine neurons were exclusively inhibited by aversive events, and expectation reduced dopamine neurons' responses to reward and punishment. When a single odor predicted both reward and punishment, dopamine neurons' responses to that odor reflected the integrated value of both outcomes. Thus, in low reward contexts, dopamine neurons signal value prediction errors (VPEs) integrating information about both reward and aversion in a common currency. In contrast, in high reward contexts, dopamine neurons acquired a short-latency excitation to aversive events that masked their VPE signaling. Our results demonstrate the importance of considering the contexts to examine the representation in dopamine neurons and uncover different modes of dopamine signaling, each of which may be adaptive for different environments.

\*For correspondence: mitsuko@mcb.harvard.edu

## Introduction

Dopamine is thought to be a key regulator of learning from appetitive as well as aversive events (*Schultz et al., 1997*; *Wenzel et al., 2015*). It has been proposed that dopamine neurons act as a teaching signal in the brain by signaling the discrepancy between the values of actual and predicted rewards, that is, reward prediction error (RPE) (*Bayer and Glimcher, 2005*; *Cohen et al., 2012*; *Hart et al., 2014*; *Roesch et al., 2007*; *Schultz, 2010*; *Schultz et al., 1997*). Although accumulating evidence supports this idea with respect to rewarding and reward-predicting events (*Bayer and Glimcher, 2005*; *Cohen et al., 2012*; *Eshel et al., 2015*; *Hart et al., 2014*; *Roesch et al., 2007*; *Schultz, 2010*; *Schultz et al., 1997*), how dopamine neurons integrate information about aversive events remains highly controversial.

Pioneering work by Wolfram Schultz and colleagues introduced the idea that dopamine neurons signal RPE. This work demonstrated that dopamine neurons in the midbrain of monkeys exhibit a highly specific set of responses to reward (*Mirenowicz and Schultz, 1994*). When the animal receives reward unexpectedly, dopamine neurons fire a burst of action potentials. If a sensory cue reliably predicts reward, however, dopamine neurons decrease their response to reward, and instead burst to the cue. Finally, if an expected reward is omitted, dopamine neurons pause their firing at the time they usually receive reward (*Hollerman and Schultz, 1998*; *Schultz et al., 1997*). Subsequently, the idea of RPE coding by dopamine neurons has been substantiated by further experiments in a variety of species including monkeys (*Bayer and Glimcher, 2005*; *Hollerman and Schultz, 1998*; *Waelti et al., 2001*), rats (*Flagel et al., 2011*; *Pan et al., 2005*; *Roesch et al., 2007*),

**eLife digest** There are many types of learning; one type of learning means that rewards and punishments can shape future behavior. Dopamine is a molecule that allows neurons in the brain to communicate with one another, and it is released in response to unexpected rewards. Most neuroscientists believe that dopamine is important to learn from the reward; however, there are different opinions about whether dopamine is important to learn from punishments or not.

Previous studies that tried to examine how dopamine activities change in response to punishment have reported different results. One of the likely reasons for the controversy is that it was difficult to measure only the activity of dopamine-releasing neurons.

To overcome this issue, Matsumoto et al. used genetically engineered mice in which shining a blue light into their brain would activate their dopamine neurons but not any other neurons. Tiny electrodes were inserted into the brains of these mice, and a blue light was used to confirm that these electrodes were recording from the dopamine-producing neurons. Specifically if the electrode detected an electrical impulse when blue light was beamed into the brain, then the recorded neuron was confirmed to be a dopamine-producing neuron.

Measuring the activities of these dopamine neurons revealed that they were indeed activated by reward but inhibited by punishment. In other words, dopamine neurons indeed can signal punishments as negative and rewards as positive on a single axis. Further experiments showed that, if the mice predicted both a reward and a punishment, the dopamine neurons could integrate information from both to direct learning.

Matsumoto et al. also saw that when mice received rewards too often, their dopamine neurons did not signal punishment correctly. These results suggest that how we feel about punishment may depend on how often we experience rewards.

In addition to learning, dopamine has also been linked to many psychiatric symptoms such as addiction and depression. The next challenge will be to examine how the frequency of rewards changes an animal's state and responses to punishment in more detail, and how this relates to normal and abnormal behaviors.

---

mice (*Cohen et al., 2012*; *Eshel et al., 2015*) and humans (*D'Ardenne et al., 2008*). This signal is proposed to underlie associative learning (*Rescorla and Wagner, 1972*), and bears a striking resemblance to machine learning algorithms (*Sutton and Barto, 1998*).

Many of the previous studies that characterized dopamine responses used rewarded outcomes with varying degrees of predictability. Comparatively fewer studies have used aversive stimuli in the context of prediction errors. Among studies that have used aversive stimuli, these provide differing reports as to how dopamine neurons respond to aversive stimuli (*Fiorillo, 2013*; *Schultz, 2015*; *Wenzel et al., 2015*).

It is thought that the majority of dopamine neurons are inhibited by aversive stimuli (*Mileykovskiy and Morales, 2011*; *Mirenowicz and Schultz, 1996*; *Tan et al., 2012*; *Ungless et al., 2004*). However, a number of electrophysiological recording studies have reported that dopamine neurons are activated by aversive stimuli both in anesthetized (*Brischoux et al., 2009*; *Coizet et al., 2006*; *Schultz and Romo, 1987*) and awake animals (*Guarraci and Kapp, 1999*; *Joshua et al., 2008*; *Matsumoto and Hikosaka, 2009*), although the proportions and locations of aversion-activated neurons differed among these studies. The differences in the results between these studies could be due to the heterogeneity of dopamine neurons or to differences in experimental conditions (e.g. type of aversive stimuli; type of anesthesia). Furthermore, another study using fast-scan cyclic voltammetry found that dopamine neurons are excited during successful avoidance of aversive stimuli (*Oleson et al., 2012*), which could be 'rewarding'. Therefore, some of the excitatory responses to aversive stimuli may not be due to aversiveness alone.

Some of these discrepancies could correspond to differences in dopamine signaling depending on the projection target. *Roitman et al. (2008)* monitored dopamine dynamics in the nucleus accumbens using cyclic voltammetry while the animal received intra-oral administrations of a sucrose or quinine solution (*Roitman et al., 2008*). This study found that these stimuli caused opposite

responses: dopamine release was increased by sucrose and decreased by quinine (*McCutcheon et al., 2012*), suggesting that at least the majority of dopamine neurons projecting to the nucleus accumbens are inhibited by aversive stimuli. *Matsumoto and Hikosaka (2009)* examined the diversity of dopamine neurons in context of prediction error. They showed that dopamine neurons that are activated by the prediction of aversive stimuli are located in the lateral part of the substantia nigra pars compacta (SNc), supporting the notion that dopamine subpopulations are spatially segregated (*Matsumoto and Hikosaka, 2009*). Consistent with this finding, *Lerner et al. (2015)* showed, using calcium imaging with fiber photometry, that SNc neurons projecting to the dorsolateral striatum are activated by aversive stimuli (electric shock) whereas those projecting to the dorsomedial striatum are inhibited (*Lerner et al., 2015*). *Lammel et al. (2011)* provided further evidence for spatial heterogeneity by showing that dopamine neurons projecting to the medial prefrontal cortex, located in the medial ventral tegmental area (VTA) exhibited a form of synaptic plasticity (AMPA/NMDA ratio) in response to aversive stimuli (formalin injection) whereas dopamine neurons projecting to the dorsolateral striatum did not (*Lammel et al., 2011*) although how these neurons change their firing patterns in response to aversive stimuli remains unknown.

In contrast to the above findings suggesting that dopamine neurons are heterogeneous with respect to signaling aversive events, Schultz, Fiorillo and colleagues have argued that dopamine neurons largely ignore aversiveness (*Fiorillo, 2013*; *Schultz, 2015*; *Stauffer et al., 2016*). One argument is that the excitation of dopamine neurons caused by aversive stimuli may be due to a 'generalization' or 'spill-over' effect of rewarding stimuli. Specifically, *Mirenowicz and Schultz (1996)* showed that when rewarding and aversive stimuli are predicted by similar cues (e.g. in a same sensory modality), aversion-predicting cues increase their tendency to activate dopamine neurons ('generalization') (*Mirenowicz and Schultz, 1996*). *Kobayashi and Schultz (2014)* showed that in a high-reward context, cues that predict a neutral outcome (e.g. a salient picture) increased their tendency to activate dopamine neurons compared to the neutral cues in a low reward context (*Kobayashi and Schultz, 2014*). Based on these and other observations (*Fiorillo et al., 2013*; *Nomoto et al., 2010*), they proposed that the early response reflects attributes such as stimulus generalization and intensity, and the later response reflects the subjective reward value and utility (*Schultz, 2016*; *Stauffer et al., 2016*).

One influential paper by *Fiorillo (2013)* concluded that dopamine neurons represent prediction errors with respect to reward but not aversiveness (*Fiorillo, 2013*). That is, dopamine neurons ignore aversive events. Recording from non-human primates, Fiorillo used three pieces of evidence to support this claim: First, dopamine neurons' responses to aversive outcomes (air puff) were indistinguishable from their responses to neutral outcomes. Second, although most dopamine neurons reduced their reward responses when the reward was predicted, their response to aversive events was unaffected by prediction. Third, dopamine neurons did not integrate the value of aversive events when combined with rewarding events. From these results, the author proposed that the brain represents reward and aversiveness independently along two dimensions (*Fiorillo, 2013*). As a result, the author proposed that different molecules regulate different types of reinforcement learning: dopamine for reward and a different molecule for aversiveness. If proven true, these ideas are fundamental in understanding how the brain learns from reward and aversion. However, it remains to be clarified whether these observations can be generalized.

The conclusions in many of the studies cited above relied upon indirect methods such as spike waveforms and firing properties (*Ungless and Grace, 2012*) in order to identify dopamine neurons. These identification methods differed among studies and have recently been called into question (*Lammel et al., 2008*; *Margolis et al., 2006*; *Ungless and Grace, 2012*). The ambiguity of cell-type identification criteria across studies makes it difficult to consolidate data on dopamine signaling. For example, Ungless et al. showed that some neurons in the VTA that were excited by aversive events and identified as dopaminergic using standard electrophysiological criteria were revealed not to be dopaminergic when they were examined with juxtacellular labeling (*Ungless et al., 2004*). Furthermore, Schultz has argued that some previous recording studies may not have targeted areas rich in dopamine neurons (*Schultz, 2016*).

To circumvent this problem, we tagged dopamine neurons with a light-gated cation channel, channelrhodopsin-2 (ChR2) and unambiguously identified dopamine neurons based on their responses to light (*Cohen et al., 2012*). In the present study, we monitored the activity of identified dopamine neurons using a series of behavioral tasks designed to determine how dopamine neurons

encode prediction of aversive events in addition to reward. Our results demonstrate that, in contrast to the proposal by *Fiorillo (2013)*, dopamine neurons in VTA indeed are able to encode complete VPE, integrating information about both appetitive and aversive events in a common currency. Importantly, the ability of dopamine neurons to encode VPE depends on both reward contexts and the animal's trial-by-trial behavioral state.

## Results

### Identification of dopamine neurons and task designs

We recorded the spiking activity of total 176 neurons in the VTA using tetrodes while mice performed classical conditioning tasks (*Table 1*). To identify neurons as dopaminergic, we optogenetically tagged dopamine neurons (*Cohen et al., 2012*). We then used a method developed previously (Stimulus-Associated spike Latency Test [SALT]) (*Eshel et al., 2015*; *Kvitsiani et al., 2013*; *Tian and Uchida, 2015*) to determine whether light pulses significantly changed a neuron's spike timing (p<0.001, *Figure 1*). To ensure that spike sorting was not contaminated by light artifacts, we compared the waveforms between spontaneous and light-evoked spikes, as described previously (*Cohen et al., 2012*). Dopamine neurons were mostly recorded from the central and posterior part of the lateral VTA including the parabrachial pigmented nucleus (PBP), parainterfascicular nucleus (PIF) and paranigral nucleus (PN) (*Figure 1G,K,O*). We obtained 72 optogenetically-identified dopamine neurons in total (5 ± 4 neurons per mouse; mean ± S.D.; *n* = 14 mice).

We devised several different tasks to characterize dopamine activities in response to mild aversive air puff (*Table 1*). 'Mixed prediction task' (low reward context) was designed to examine interaction between the prediction of reward and the prediction of aversiveness. 'Low reward probability task' (low reward context) and 'high reward probability task' (high reward context) were specifically designed to test the effects of reward probability on dopamine responses: two task conditions differed only with respect to reward probabilities. 'High reward probability task 2' (high reward context) was originally conducted to replicate the diverse responses to aversive stimuli in dopamine neurons, which were reported in multiple previous studies. The effects of reward contexts were also examined with the mixed prediction task and the high reward probability task 2.

In the present study, we first focused on the characterization of dopamine activities in low reward contexts (*Figures 2–4*). Then, we compared dopamine activities between low and high reward contexts (*Figure 5*). Finally, we examined dopamine activities in relation to behaviors in different contexts (*Figure 6*).

### Dopamine neurons integrate values of both valences, appetitive and aversive

A previous study reported that dopamine neurons do not integrate information about aversiveness along with reward-related information when rewarding liquid and an air puff are delivered to a

**Table 1.** Summary of task conditions.

| Task | Outcome | CS (% outcome) | | | | Reward trials (%) | Free reward (%) |
|---|---|---|---|---|---|---|---|
| | | Odor A (Reward CS) | Odor B (Nothing CS) | Odor C (Air puff CS) | Odor D (Reward and air puff CS) | | |
| Mixed prediction task | Water | 25 | 0 | 0 | 25 | 13 | 2 |
| | Air puff | 0 | 0 | 75 | 75 | | |
| Low reward probability task | Water | 20 | 0 | 0 | | 7 | 6 |
| | Air puff | 0 | 0 | 90 | | | |
| High reward probability task | Water | 90 | 0 | 0 | | 30 | 6 |
| | Air puff | 0 | 0 | 90 | | | |
| High reward probability task 2 | Water | 90 | 0 | 0 | | 30 | 7 |
| | Air puff | 0 | 0 | 80 | | | |

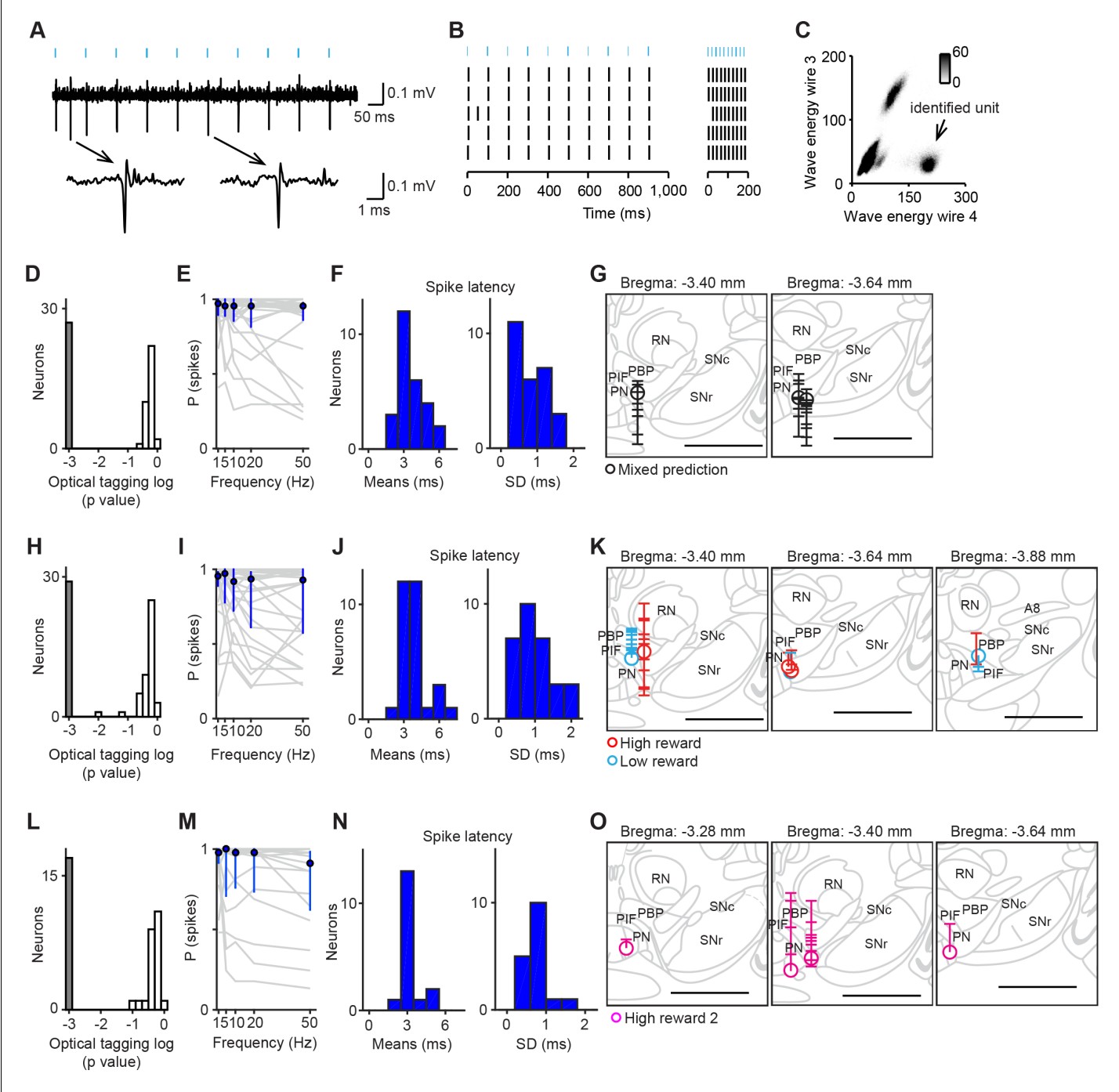

**Figure 1.** Optogenetic identification of dopamine neurons in the ventral tegmental area (VTA). (**A**) Voltage trace from 10 pulses of 10 Hz light stimulation (cyan bars, top) of a representative dopamine neuron. A spontaneous spike and a light-triggered spike were magnified at the bottom. (**B**) Responses from this neuron to 10 Hz (left) and 50 Hz (right) stimulation. (**C**) Isolation of an identified dopamine neuron from noise and other units. (**D**) Histogram of p values testing whether light-activation induced significant changes in spike timing ($n$ = 62 neurons) in the mixed prediction task. The p values were derived from SALT (Stimulus-Associated spike Latency Test; see Materials and methods). Neurons with p values < 0.001 and waveform correlations > 0.9 were considered identified (grey). P values and waveform correlations were calculated using light stimulation with all the frequencies (1–50 Hz). (**E**) Probability of light-evoked spike as a function of stimulation frequency for each dopamine neuron (grey) and the average across dopamine neurons (blue circles and bars, median and interquartile range). (**F**) Histograms of the mean (left) and S.D. (right) spike latency to light stimulation with all the frequencies (1–50 Hz) for 26 identified dopamine neurons. (**G**) Reconstruction of the positions of individual dopamine neurons recorded in the mixed prediction task. Each circle represents a lesion site from individual animals used in the mixed prediction task. Each horizontal line on the track (indicated by a vertical line over the lesion site) indicates estimated recording positions of individual dopamine neurons. Labeled

*Figure 1 continued on next page*

*Figure 1 continued*

structures: parabrachial pigmented nucleus of the VTA (PBP), parainterfascicular nucleus of the VTA (PIF), paranigral nucleus of the VTA (PN), red nucleus (RN), substantia nigra pars compacta (SNc), and substantia nigra pars reticulata (SNr). Scale bar, 1 mm. (H–J) Optogenetic identification of dopamine neurons recorded in high and low reward probability tasks (29 dopamine neurons identified out of 73 neurons). Conventions are the same as in D–F. (K) Reconstruction of the positions of individual dopamine neurons recorded in high (red) and low (cyan) reward probability tasks. Conventions are the same as in G. (L–N) Optogenetic identification of dopamine neurons recorded in high reward probability task 2 (17 dopamine neurons identified out of 41 neurons). (O) Reconstruction of the positions of individual dopamine neurons recorded in high reward probability task 2 (magenta).

monkey at the same time (*Fiorillo, 2013*). However, this method may produce complex interactions between the two different outcomes. To test how reward and aversion interact and affect dopamine responses, we devised a 'mixed prediction' paradigm (*Figure 2*) in which a single odor (Odor D in *Figure 2A*, conditioned stimulus, CS) predicted both a rewarding and a mildly aversive event in a complementary and probabilistic manner: a reward (water) was delivered in 25% of the trials and an aversive event (air puff) was delivered in the remaining 75% of the trials. For comparison, we included the following trial types: Odor A predicted water in 25% of trials (nothing in 75%), Odor C predicted air puff in 75% of trials (nothing in 25%), and Odor B predicted no outcome. Each behavioral trial began with the odor CS (1 s), followed by a 1-s delay and an unconditioned stimulus (US). We chose higher probability for air puff than water in order to balance the strength of positive and negative values in the task; we suspect that the magnitude of the negative value of mild air puff is much smaller than the magnitude of the positive value of water, which could cause us to overlook a small effect of predicted air puff on the CS response.

We first asked whether the recorded dopamine neurons were inhibited or excited by odor cues (CSs) that predicted different outcomes. We found that the vast majority of the neurons were inhibited by the air puff-predicting CS while excited by the reward-predicting CS (*Figure 2B–D*). On average, the firing rate during the CS period was significantly lower for the air puff-predicting CS than for the CS predicting nothing, while it was higher for the reward-predicting CS than for the CS that predicted nothing (*Figure 2E*). A similar tendency was observed using data from two animals instead of three (i.e. leaving one animal out of three) (*Figure 2—figure supplement 1*). Among 26 identified dopamine neurons, 85% (22 neurons) were significantly modulated by these three odors (p<0.05, one-way ANOVA), and 59% (13 of 22 significant neurons) showed the monotonic CS value coding (water > nothing > air puff). These results suggest that the firing of identified dopamine neurons was negatively modulated by the stimulus predicting aversive events.

We next examined whether prediction of aversion in addition to reward changed the response of dopamine neurons. In contrast to the previous study (*Fiorillo, 2013*), we found that the majority of neurons showed an intermediate response to the CS predicting both water and air puff (Odor D) compared to the CSs predicting water only (Odor A) or air puff only (Odor C) (*Figure 2B,F,G*). As a population, the net response to these CSs increased monotonically according to the values of both water and air puff, with the CS response to Odor D falling in between that of Odor A and Odor C (*Figure 2H*). 89% (23 of 26 neurons) of identified dopamine neurons were significantly modulated by these three odors (p<0.05, one-way ANOVA), and 65% (15 of 23 significant neurons) showed the monotonic CS value coding (water > water and air puff > air puff). These results indicate that VTA dopamine neurons combine values for both reward and punishment along a one-dimensional value axis.

## Dopamine neurons signal prediction errors for aversion

It has been shown that dopamine neurons' responses to reward are greatly reduced when the reward is predicted, a signature of prediction error coding (*Schultz et al., 1997*). We replicated these findings here even in low reward probability conditions (20–25%, *Figure 3—figure supplement 1*; see Materials and methods). We next examined whether these dopamine neurons show prediction error coding for aversive events. To address this question, we occasionally delivered air puff during inter-trial intervals without any predicting cues. These responses to unpredicted air puff were compared to the responses to air puff in trials when air puff was predicted by an odor cue. We found that the inhibitory response to an air puff was significantly reduced when the air puff was predicted by an odor cue (*Figure 3A–D*). To further examine whether dopamine neurons showed prediction

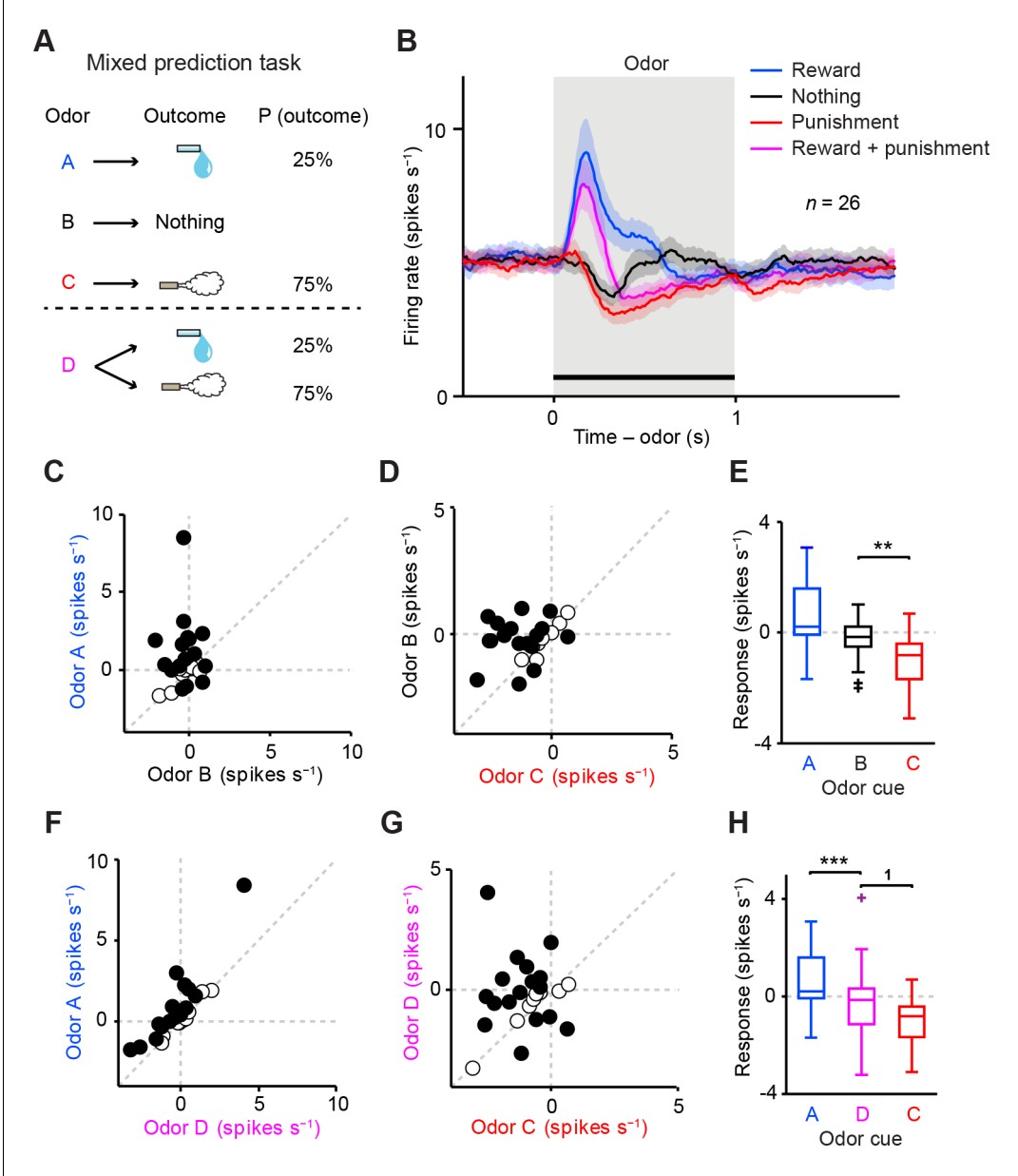

**Figure 2.** Dopamine neurons integrate values of both valences, reward and aversion. (**A**) Task design in the mixed prediction task. (**B**) Mean ± S.E.M. of firing rates of optogenetically-identified dopamine neurons during all four trial conditions; reward (blue), nothing (black), punishment (red), and both reward and punishment (magenta). (**C**) Scatter plot of the mean responses during the CS epoch (0–1 s, indicated by a solid black line in **B**) for reward versus nothing. The baseline firing rate (−1–0 s from odor onset) was subtracted for each neuron. Black filled circles indicate neurons with significant difference between responses to the CS predicting reward and that predicting nothing (unpaired $t$ test, $p<0.05$). (**D**) Scatter plot of the mean responses during the CS epoch for nothing versus punishment. Black filled circles indicate neurons with significant difference between responses to the CS predicting nothing and that predicting punishment. (**E**) Comparison of the responses of individual neurons ($n = 26$) during CS (0–1 s) predicting reward (blue), nothing (black) and punishment (red). For all box plots, the central mark is the median, the edges of the box are the 25th and 75th percentiles, and the whiskers extend to the most extreme data points not considered outliers (points 1.5 × interquartile range away from the 25th or 75th percentile), and outliers are plotted individually as plus symbols. **$t(25) = 3.7$, $p=0.001$, paired $t$ test. One outlier >5 Hz in response to Odor A is not represented. (**F**) Scatter plot of the mean responses during the CS epoch for reward versus reward and punishment. Black filled circles indicate neurons with significant difference between responses to the CS predicting reward and that predicting reward and punishment. (**G**) Scatter plot of the mean responses during the CS epoch for reward and punishment versus punishment. Black filled circles indicate neurons

*Figure 2 continued on next page*

*Figure 2 continued*

with significant difference between responses to the CS predicting punishment and that predicting reward and punishment. (H) Comparison of the responses during CS predicting reward (blue), both reward and punishment (magenta), and punishment (red). [1]$t(25)$ = 2.5, p=0.02; ***$t(25)$ = 4.4, p=2.0 $\times$ 10$^{-4}$, paired $t$ test. One outlier >5 Hz in response to Odor A is not represented.

The following figure supplement is available for figure 2:

**Figure supplement 1.** Comparison of CS responses using dopamine neurons from two animals instead of three.

---

error coding for aversive events, we compared the firing rate during the outcome period in air puff omission trials with that in trials that predict nothing. We found that the omission of a predicted air puff slightly but significantly increased firing rates, compared to no change in nothing trials (*Figure 3E–H*) although we observed variability in air puff omission responses. Together, these results demonstrate that dopamine neurons signal prediction errors for aversive events in addition to rewarding events. These results indicate that dopamine neurons have the ability to signal VPEs for both appetitive and aversive events, supporting previous work by Matsumoto and Hikosaka (*Matsumoto and Hikosaka, 2009*) and contrasting with previous work by Fiorillo (*Fiorillo, 2013*).

## Homogeneous response function of dopamine neurons

Although we found that most dopamine neurons were inhibited by air puff (mildly aversive event), there was a considerable variability in the extent to which individual dopamine neurons were inhibited. Does this diversity support a functional diversity across dopamine neurons in the lateral VTA?

In a previous study, dopamine neurons in the lateral VTA exhibited neuron-to-neuron variability in the magnitude of response to a given size of reward (*Eshel et al., 2016*). Despite this variability in responsivity, the response functions of individual dopamine neurons were scaled versions of each other, indicating a remarkable homogeneity. One consequence of this scaled relationship is that neurons that responded strongly to a given size of reward were more greatly suppressed by reward expectation. In other words, reward expectation suppressed a neuron's reward response in proportion to the size of its response to unexpected reward.

Does the same relationship hold for inhibitory responses to air puff? To address this question, we examined the correlation between aversion-related responses in dopamine neurons (*Figure 4*, *Figure 4—figure supplement 1*). We indeed found a similar relationship: dopamine neurons that were strongly inhibited by air puff also exhibited a larger prediction-dependent reduction of their responses to air puff (*Figure 4A*, Pearson's $r$ = 0.69, p=1.9 $\times$ 10$^{-6}$). In other words, the ratio between individual dopamine neurons' responses to unpredicted versus predicted air puff was preserved across neurons. In addition, similar to reward responses (*Eshel et al., 2016*), inhibitory responses to the air puff-predicting CS were correlated with prediction-dependent reduction of responses to air puff US (*Figure 4B*, Pearson's $r$ = 0.39, p=0.016). These results indicate that the response function was preserved across dopamine neurons in the case of aversive stimuli.

We next examined the relationship between responses of dopamine neurons to reward and to aversion. We compared responses of dopamine neurons to unpredicted water and unpredicted air puff (*Figure 4C*). We observed no obvious unique clusters across neurons, supporting the notion that there was no clear subpopulation of dopamine neurons specialized in signaling reward versus aversion in the lateral VTA. Rather, we found that most of dopamine neurons were inhibited by unpredicted aversive stimuli and excited by unpredicted rewarding stimuli. Interestingly, we did not find any negative or positive correlation of neurons' responses to water and air puff; the proportion of the response magnitudes in response to reward versus aversion was diverse across neurons. These results indicate that although the response function either for reward prediction error or for aversion prediction error was homogeneous across dopamine neurons, these two functions were relatively independent, suggesting that different mechanisms may produce dopamine responses to reward and aversion.

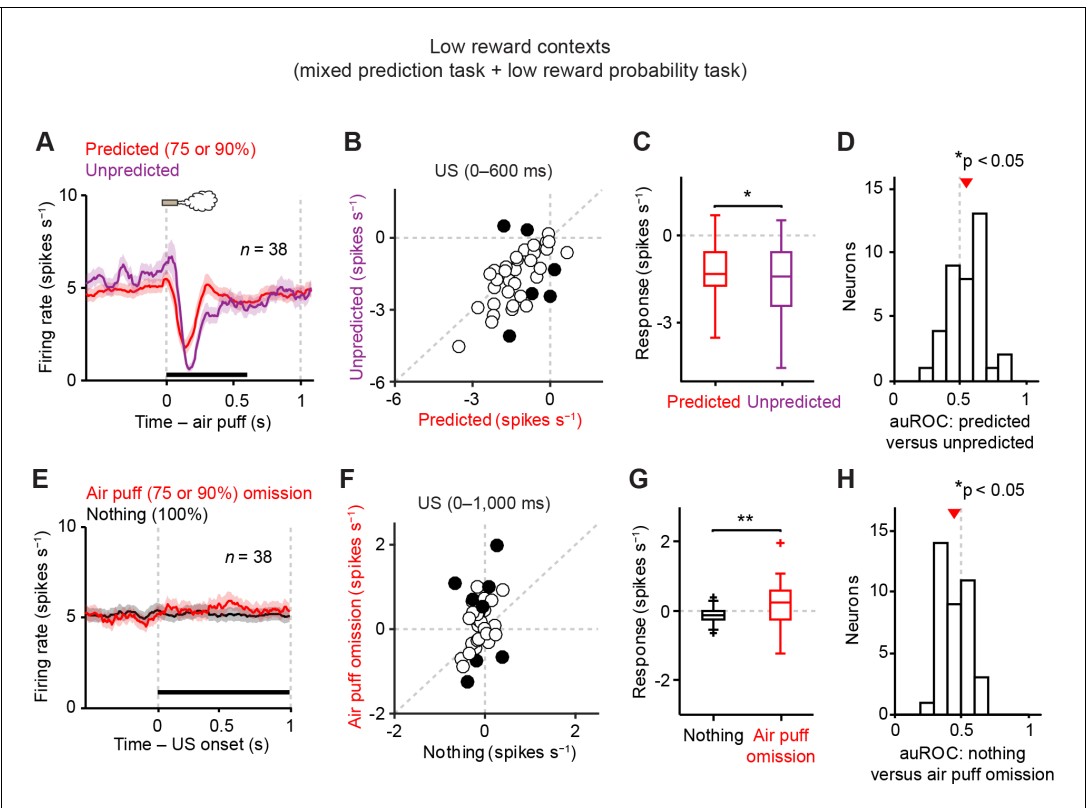

**Figure 3.** Dopamine neurons signal aversive prediction error. (**A**) Mean ± S.E.M. of firing rate of dopamine neurons in response to predicted (red) and unpredicted air puff (purple). (**B**) Scatter plot of the responses to predicted and unpredicted air puff (0–600 ms after air puff, indicated by a black solid line in **A**). Each data point represents an individual dopamine neuron. The baseline firing rate (−1–0 s from odor onset) was subtracted for each neuron. Black filled circles indicate neurons with significant difference between responses to unpredicted and predicted air puff (unpaired t test, p<0.05). (**C**) Comparison of the responses to predicted and unpredicted air puff (n = 38). For all box plots, central mark is the median, box edges are 25th and 75th percentiles, whiskers extend to the most extreme data points not considered outliers (points 1.5 × interquartile range away from the 25th or 75th percentile), and outliers are plotted as plus symbols. *t(37) = 2.4, p=0.02, paired t test. (**D**) Histogram of changes in firing rate during the US epoch (0–600 ms) of predicted versus unpredicted air puff. The population average of the auROC curve was significantly different from 0.5 (n = 38, *t(37) = 2.5, p=0.015, one-sample t test). Red arrow indicates mean auROC value. (**E**) Mean ± S.E.M. of firing rate around the outcome period in air puff omission (red) and nothing (black) trials. (**F**) A scatter plot of the firing rate during the outcome period in air puff omission trials and nothing trials (0–1000 ms after US onset, indicated by a black solid line in **E**) subtracted by the baseline firing rate (−1–0 s from odor onset) for each neuron. Black filled circles indicate neurons with significant difference between firing rates during the outcome period in air puff omitted and nothing trials. (**G**) Comparison of the responses during air puff omission trial and nothing trial conditions. **t(37) = 2.8, p=0.008, paired t test. (**H**) Histogram of changes in firing rate during the US epoch (0–1000 ms) of air puff omission versus nothing trials. The population average of auROC curve was significantly different from 0.5 (n = 38, *t(37) = 2.7, p=0.011, one-sample t test).

The following figure supplement is available for figure 3:

**Figure supplement 1.** Reward prediction error coding by dopamine neurons in low reward contexts.

## Reward-context dependent representation of aversion in dopamine neurons

Although the above results suggested that most of the dopamine neurons that we recorded from the lateral VTA were inhibited by aversive events, contrasting results were obtained in some previous studies. In monkeys, it was found that, on average, the responses to aversive stimuli were indistinguishable from responses evoked by neutral stimuli (*Fiorillo, 2013*). In addition, previous studies in mice (*Cohen et al., 2012*; *Tian and Uchida, 2015*) mirrored these contrasting results in VTA dopamine neurons. These results suggest that the difference between studies is not due to a species difference, raising the possibility that our task parameters altered the dopamine response.

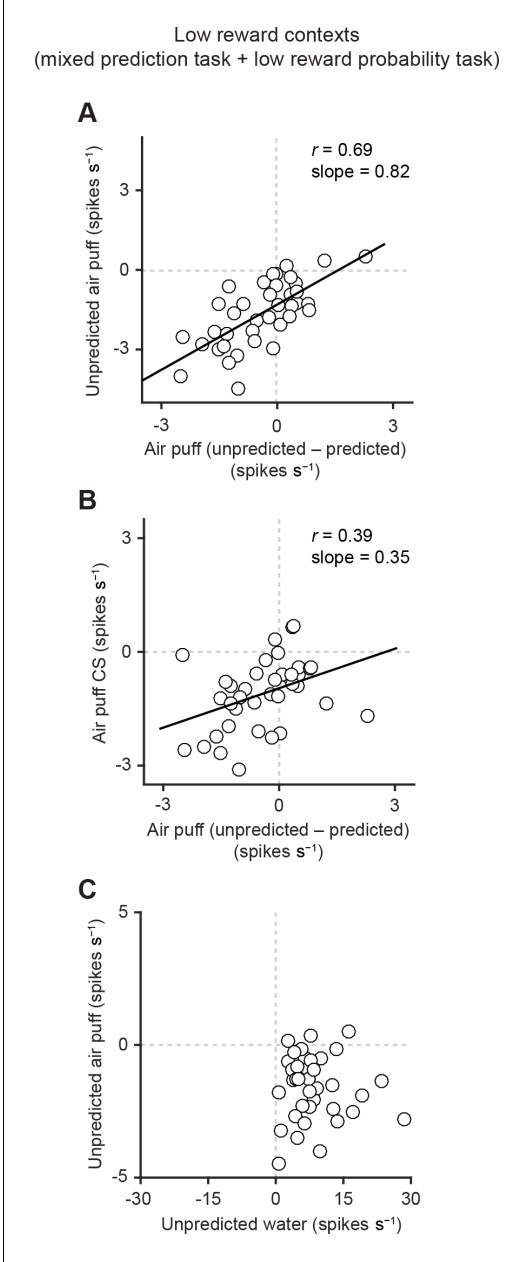

**Figure 4.** Correlation between responses related to aversive stimuli in dopamine neurons. (**A**) Scatter plot of the responses to unpredicted air puff (0–600 ms from air puff onset) and the effects of prediction on the responses to air puff US (subtraction of responses to unpredicted air puff from responses to predicted air puff, 0–600 ms from air puff onset) in dopamine neurons. Each data point represents an individual dopamine neuron ($n = 38$). The baseline firing rate ($-1$– 0 s from odor onset) was subtracted for each neuron. Solid line, best-fit linear regression. Pearson's correlation, $r = 0.69$, $p = 1.9 \times 10^{-6}$. (**B**) Scatter plot of the responses to CS predicting air puff (0–1000 ms from odor onset) and the effects of prediction on the responses to air puff in dopamine neurons ($n = 38$).

*Figure 4 continued on next page*

Multiple studies found that even non-reward-ing stimuli can excite dopamine neurons with short latency. A recent study reported that whether a neutral stimulus elicits these short-latency excitations depends on reward context, and that excitation is larger in a high versus a low reward context (*Kobayashi and Schultz, 2014*). We noted that the reward probability used in the task described above (mixed prediction task) was much lower (15% rewarded trials overall) than in previous studies (e.g. 50% rewarded trials overall in *Cohen et al., 2012*). This raises the possibility that in a high reward context, short-latency exci-tations to aversive stimuli masked inhibitory responses to aversive stimuli, and thus clear inhi-bition by aversive stimuli has not been observed in previous studies because these studies typi-cally used relatively high reward probabilities.

To directly test whether reward probability affected dopamine neurons' responses to aver-sive events, we recorded the activity of dopamine neurons in two task conditions that differed only with respect to reward probabilities (*Figure 5*). In the high reward probability condition, the proba-bility of water in Odor A trials was 90% (36% reward trials overall) (*Figure 5A*) while in the low reward probability condition, the reward proba-bility in Odor A trials was 20% (13% reward trials overall) (*Figure 5F*). Consistent with the previous study (*Kobayashi and Schultz, 2014*), nothing-predicting CS (neutral cue) elicited short-latency excitation more prominently in the high reward compared to the low reward probability condi-tion (*Figure 5B,G*). Next we examined the response to the air puff-predicting CS. In the low reward context, the difference between the responses to air puff- and nothing-predicting CSs remained significantly different (*Figure 5H–J*), consistent with the above experiment (*Figure 2*). In contrast, we found that in the high reward probability condition, dopamine neurons exhib-ited biphasic responses to the air puff-predicting CS: short-latency excitation followed by later inhi-bition. Furthermore, dopamine neurons' net responses to the air puff-predicting CS and noth-ing-predicting CS were no longer significantly dif-ferent (*Figure 5C–E*). This result, obtained in a high reward probability condition, is similar to those obtained in previous studies (*Cohen et al., 2012*; *Fiorillo, 2013*; *Tian and Uchida, 2015*). In the low reward context, all the dopamine neurons (12 of 12 dopamine neurons; $p < 0.05$, one-sample $t$ test; filled grey circles in *Figure 5H*) were inhib-ited by the air puff-predicting CS whereas in the high reward context, a large fraction of dopamine neurons (6 of 17 dopamine neurons; $p > 0.05$,

*Figure 4 continued*

Pearson's correlation, r = 0.39, p=0.016. (**C**) Scatter plot of the responses of individual dopamine neurons (*n* = 37) to unpredicted water and air puff (0–600 ms from water and air puff onsets, respectively). No correlation between these two responses (Pearson's correlation, *r* = -0.04, p=0.834).

The following figure supplement is available for figure 4:

**Figure supplement 1.** Correlation between responses related to air puff omission and the effects of prediction on the responses to air puff US.

one-sample *t* test; filled white circles in *Figure 5C*) did not show consistent inhibition by air puff CS. That is, in the high reward context, a little fraction of neurons showed a stronger inhibition to the air puff CS compared to the nothing CS (100% and 65%, low and high reward context, respectively; p=0.02, chi-square test; *Figure 5K*).

We obtained additional data using a task condition similar to the high reward context (high reward probability task 2; see Materials and methods, and *Table 1*) (*n* = 17 identified dopamine neurons). Furthermore, the mixed prediction task (*Figure 2*) provides additional data for a low reward context (*n* = 26 identified dopamine neurons). Similar results were obtained by using dopamine neurons in each of these experiments or by pooling neurons from these experiments separately for high and low reward contexts (*n* = 34 and 38 identified dopamine neurons, respectively) (*Figure 5—figure supplement 1*).

The above analyses used a relatively large time window that contains the entire response period (0–1,000 ms). Because dopamine responses in high reward contexts exhibited biphasic responses (early excitation followed by later inhibition), we further analyzed the data by separating these time windows into smaller bins. Because there is no known mechanism by which downstream neurons can read out these windows separately, analysis using a large window can be considered more conservative. However, previous studies have proposed that different information may be conveyed in these time windows (*Schultz, 2016*; *Stauffer et al., 2016*).

We obtained similar results even if we compared only later time bins (200–1,000 ms), excluding the early excitation phase (*Figure 5—figure supplement 2*). By excluding the early excitation period (0–200 ms), many dopamine neurons showed inhibition to air puff-predicting CS in both low and high reward contexts compared to the baseline firing (92% and 82%, respectively). However, during this inhibition phase, most dopamine neurons (65%) did not distinguish the air puff CS from the nothing CS in the high reward context while most dopamine neurons (75%) showed more inhibition to the air puff CS than to the nothing CS in the low reward context (i.e. 35% and 75% of neurons distinguished air puff CS from nothing CS in high and low reward contexts, respectively; p=0.04, chi-square test; *Figure 5—figure supplement 2C*). This suggests that although many dopamine neurons exhibit inhibitory responses during the later response window in high reward contexts, the information that this inhibition conveys may be different from that in low reward contexts; the inhibition in the high reward context largely reflected 'no reward' rather than the negative value of an air puff (*Fiorillo, 2013*). Neurons that showed a large excitation to the air puff CS were not necessarily the same group of neurons which showed excitation to the air puff itself, consistent with a previous study (*Matsumoto and Hikosaka, 2009*) (*Figure 5—figure supplement 3*).

These results demonstrate that dopamine neurons' responses to aversion-predicting cues are greatly affected by reward contexts, and suggest that dopamine neurons' ability to faithfully represent negative values of aversive cues are undermined in high reward contexts.

## Trial-to-trial variability in dopamine responses to aversive stimuli

In order to examine dopamine responses to aversive stimuli more carefully in relation to behavior, we characterized dopamine activities and anticipatory eye-blinking on a trial-by-trial basis. We quantified the area of the right eye (including both sclera and pupil) concurrently with neuronal recording in high and low reward probability contexts (*Figure 6*; *Figure 6—figure supplement 1*; see also Materials and methods). We observed that the eye area became smaller after the onset of a CS predicting air puff and became larger after a CS predicting reward (*Figure 6A,B*). In air puff trials, the eye area during the delay period was significantly smaller than before CS presentation (−1–0 s from CS onset), indicating anticipatory eye-blinking (*Figure 6C*, n = 21 sessions, p=3.4 × $10^{-12}$, paired *t* test). We confirmed that in both low and high reward probability conditions, all of the animals showed significant anticipatory eye-blinking (*Figure 6—figure supplement 2*). The eye area during

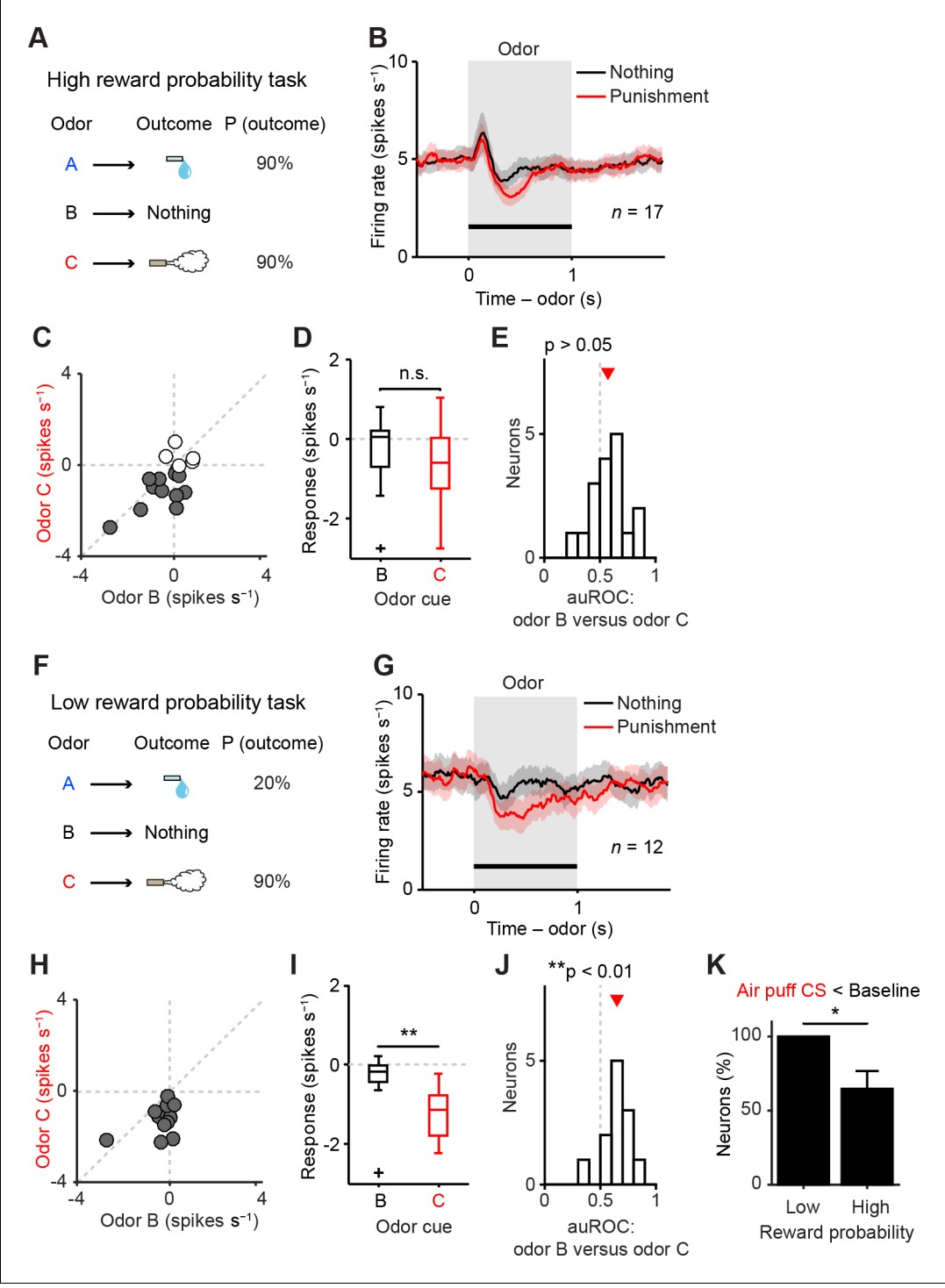

**Figure 5.** Representation of negative value of aversive stimuli depends on reward context. (**A**) Task design in the high reward probability task. (**B**) Mean ± S.E.M. of firing rate of optogenetically-identified dopamine neurons during two trial conditions; nothing (black) and punishment (red). (**C**) Scatter plot of the mean responses during the CS epoch (0–1 s, indicated by a solid black line in **B**) for punishment versus nothing. The baseline firing rate (−1–0 s from odor onset) was subtracted for each neuron. Filled grey circles (11 out of 17 circles), dopamine neurons showing significant inhibition to punishment CS than the average baseline firing rate (n = 80 trials, p<0.05, one-sample t test). (**D**) Comparison of the responses of individual neurons during CS (0–1 s) predicting nothing (black) and punishment (red). For all box plots, central mark is the median, box edges are 25th and 75th percentiles, whiskers extend to the most extreme data points not considered outliers (points 1.5 × interquartile

*Figure 5 continued on next page*

*Figure 5 continued*

range away from the 25th or 75th percentile), and outliers are plotted as plus symbols. $t(16)$ = 2.1, p>0.05, paired $t$ test. n.s., not significant. (E) Histogram of changes in firing rate during the CS epoch (0–1 s) of nothing versus punishment. The population average of the area under the receiver-operating characteristic (auROC) curve was not significantly different from 0.5 ($n$ = 17, $t(16)$ = 1.9, p>0.05, one-sample $t$ test). Red arrow indicates mean auROC value. (F) Task design in the low reward probability task. (G) Mean ± S.E.M. of firing rate of optogenetically-identified dopamine neurons during two trial conditions; nothing (black) and punishment (red). (H) Scatter plot of the mean responses during the CS epoch (0–1 s, indicated by a solid black line in G) for punishment versus nothing. Filled grey circles (12 out of 12 circles), dopamine neurons showing significant inhibition to punishment CS than the average baseline firing rate ($n$ = 80 trials, p<0.05, one-sample $t$ test). (I) Comparison of the responses of individual neurons during CS (0–1 s) predicting nothing (black) and punishment (red). **$t(11)$ = 3.8, p=0.003, paired $t$ test. (J) Histogram of changes in firing rate during the CS epoch (0–1 s) of nothing versus punishment. The population average of the auROC curve was significantly different from 0.5 ($n$ = 12, **$t(11)$ = 4.2, p=0.002, one-sample $t$ test). (K) Comparison of the percentage of dopamine neurons showing significant inhibition to punishment CS than baseline (Air puff CS < Baseline) between high and low reward probability tasks. *$chi(1)$ = 5.34, p=0.02, chi-square test. Error bar, S.E.M.

The following figure supplements are available for figure 5:

**Figure supplement 1.** The response to air puff-predicting CS was consistently smaller than that to nothing-predicting CS in low reward contexts.

**Figure supplement 2.** The later response to air puff-predicting CS was smaller than that to nothing-predicting CS in the low reward probability task.

**Figure supplement 3.** Scatter plot of responses of individual dopamine neurons to air puff-predicting CS and unpredicted air puff in high reward contexts.

the delay period was significantly smaller in air puff trials than in nothing trials (*Figure 6—figure supplement 2I*). These results indicate that our air puff conditions were aversive enough to cause anticipatory eye-blinking during the recording experiments, although we noticed that the amount of eye-blinking differed across trials.

Because the level of anticipatory eye-blinking varied across trials, we next divided air puff trials in each session into two groups, 'blink' (small eye size) and 'no-blink' (big eye size) trials (see Materials and methods), and then examined the correlation between blinking and the responses of dopamine neurons to the air puff CS. We found that the firing rates of dopamine neurons to air puff CS in blink trials were significantly smaller than that in no-blink trials (*Figure 6D*). In other words, inhibition of dopamine neurons during the CS period, but not excitation, predicted aversion-related behavior.

The correlation between trial-by-trial dopamine activity and anticipatory blinking was even clearer if we consider reward contexts (*Figure 6E*). In the low reward probability condition, the inhibitory response of dopamine neurons in blink trials was significantly greater than in no-blink trials (p=0.02, paired $t$ test). Of note, in the high reward probability condition, the inhibitory response of dopamine neurons was greatly reduced even when the animals showed anticipatory eye-blinking (*Figure 6E*, p=0.009, unpaired $t$ test). This result suggests that dopamine responses may not directly trigger eye-blinking behavior. Rather, the results are consistent with the idea that dopamine neurons' inhibitory responses to aversive cues signal negative values of the outcome, but not the action itself. Importantly, dopamine neurons showed significant inhibition only when animals showed anticipatory eye-blinking in the low reward context (*Figure 6E*, p=4.3 $\times$ $10^{-5}$, one-sample $t$ test). The results do not change even when we only used a later window of dopamine CS responses, excluding the early excitation period (0–200 ms) (*Figure 6—figure supplement 3*).

Whereas mice showed anticipatory blinking in 48% of air puff trials (90% air puff trials), they showed more consistent anticipatory licking in water trials (98% in 90% water trials) (see Materials and methods). This could be due to the fact that we used only a mildly aversive air puff (to prevent the animal from being discouraged to perform the task altogether), whereas water is highly rewarding. Although the number of trials in which the animal did not show anticipatory licking was small, we observed a similar relationship between dopamine responses and behavior: dopamine neurons

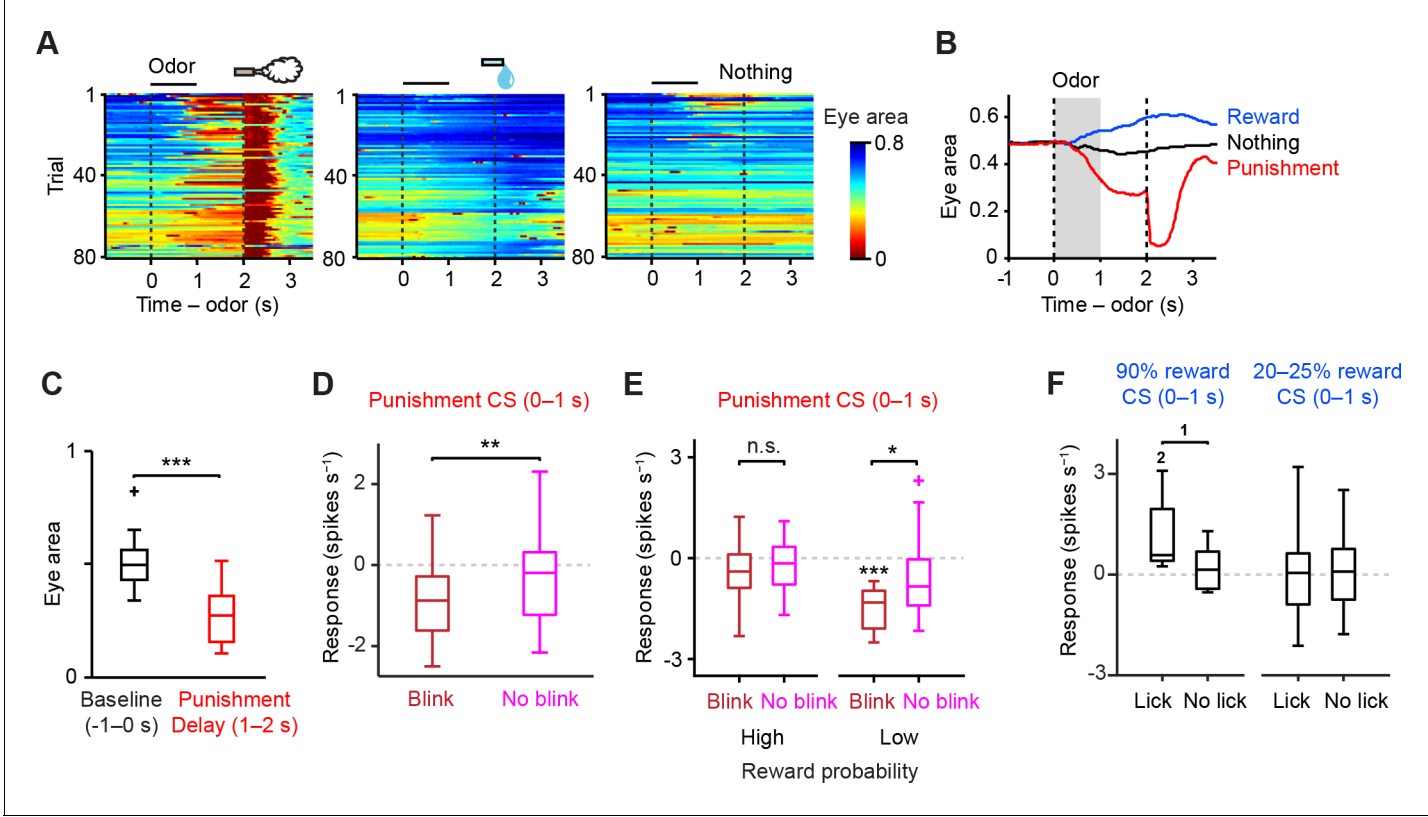

**Figure 6.** Relation between CS response of dopamine neurons and behavior. (**A**) Eye blinking behavior during all three trial conditions in an example session in the high reward probability task. Red color indicates small eye area. (**B**) Average eye area during all three trial conditions in the example session; reward (blue), nothing (black) and punishment (air puff, red). (**C**) Comparison of the eye area during baseline (−1–0 s from odor onset, black) and delay period (1–2 s, red) in punishment trial condition from an example animal (*n* = 21 sessions). For all box plots, central mark is the median, box edges are 25th and 75th percentiles, whiskers extend to the most extreme data points not considered outliers (points 1.5 × interquartile range away from the 25th or 75th percentile), and outliers are plotted as plus symbols. ***$t(20)$ = 14.7, p=3.4 × $10^{-12}$, paired *t* test. (**D**) Comparison of the responses of individual dopamine neurons (*n* = 23) during punishment CS (0–1 s) between blink (dark red) and no-blink trials (magenta). The baseline firing rate (−1–0 s from odor onset) was subtracted for each neuron. **$t(22)$ = 3.0, p=0.007, paired *t* test. (**E**) Comparison of the responses of individual dopamine neurons during punishment CS (0–1 s) in blink and no-blink trials in high and low reward probability tasks (*n* = 13 and 10 neurons from these two tasks, respectively). The baseline firing rate (−1–0 s from odor onset) was subtracted for each neuron. ***$t(9)$ = 7.4, p=4.3 × $10^{-5}$, one-sample *t* test; *$t(9)$ = 3.0, p=0.016; and $t(12)$ = 1.3, p=0.207, paired *t* test. n.s., not significant. (**F**) Comparison of the responses of individual dopamine neurons during reward-predicting CS (0–1 s) between trials with anticipatory and no anticipatory licks (≥3 and <3 licks $s^{-1}$ during delay period, respectively). In 90% water trials (left), only 9 out of 34 dopamine neurons were collected, as the number of trials in which the animal did not show anticipatory licking was small. In 20–25% water trials (right), 33 out of 38 dopamine neurons were collected. The baseline firing rate (−1–0 s from odor onset) was subtracted for each neuron. [1]$t(8)$ = 2.322, p=0.0488, paired *t* test; [2]$t(8)$ = 3.141, p=0.0138, one-sample *t* test.

The following figure supplements are available for figure 6:

**Figure supplement 1.** Extraction of eye area from video frames.

**Figure supplement 2.** All the animals showed anticipatory eye-blinking to air puff in both low and high reward probability conditions.

**Figure supplement 3.** Correlation between eye-blinking behavior and inhibitory responses to air puff-predicting CS during the later response window in low reward contexts.

**Figure supplement 4.** Anticipated licking behavior during delay period.

were consistently excited by the 90% reward cue when they showed anticipatory licking, but not when they did not show anticipatory licking (*Figure 6F*). These results indicate the importance of both reward contexts and behavioral outcomes to understand how dopamine neurons represent reward and aversion. Without monitoring behaviors, investigators may easily miss weak inhibitory responses to mildly aversive stimuli in dopamine neurons.

## Discussion

There have been divergent and inconsistent results with respect to how dopamine neurons encode aversive events. In the present study, we aimed to specifically examine how dopamine neurons integrate aversiveness and reward by carefully controlling various experimental parameters. The following three points are particularly notable. First, we used an optogenetic tagging method to unambiguously identify dopamine neurons while recording spiking activity in behaving mice. Second, we used a series of behavioral paradigms with probabilistic outcomes that are designed to test specific hypotheses regarding the integration of different outcomes and the effect of reward contexts. Third, we monitored aversion-related behaviors (eye blinking) to examine the trial-by-trial relationship to dopamine responses. By harnessing these controlled experimental conditions, our results indicate that dopamine neurons have different modes of signaling. In low reward contexts (the mixed prediction task and the low reward probability task), dopamine neurons were inhibited by aversive events more strongly than by neutral events, tracking the aversiveness of stimuli on a trial-by-trial basis. Furthermore, dopamine neurons signaled VPEs by faithfully combining information about both appetitive and aversive events into a common currency of value. Thus, dopamine can function as a precise teaching signal for updating integrated values for both reward and aversion. However, in high reward contexts (the high reward probability task and the high reward probability task 2), dopamine neurons exhibited early excitations, which undermined their ability to produce negative responses for aversive events.

### Integration of aversiveness and reward in dopamine neurons

Dopamine has long been thought to be a key regulator of reinforcement learning. One dominant theory posits that dopamine acts as a teaching signal that broadcasts an RPE signal to the rest of the brain. Recent studies using optogenetics have established that activation of dopamine neurons alone is sufficient for appetitive conditioning (*Steinberg et al., 2013*; *Tsai et al., 2009*; *Witten et al., 2011*) whereas suppression is sufficient for aversive conditioning (*Chang et al., 2016*; *Danjo et al., 2014*; *Ilango et al., 2014*; *Tan et al., 2012*; *van Zessen et al., 2012*), although activation of dopamine neurons that project to the cortex or dopamine neurons in the dorsal raphe has potential to induce aversion and/or other functions (*Lammel et al., 2014*; *Matthews et al., 2016*; *Popescu et al., 2016*). Furthermore, pharmacological studies have suggested that normal dopamine signaling is required for appetitive as well as aversive conditioning (*Cooper et al., 1974*; *Flagel et al., 2011*; *Wenzel et al., 2015*). These results have provided convergent evidence supporting the role of dopamine in learning. However, whether dopamine neurons signal prediction errors with respect to aversive events remained controversial, and remained an obstacle towards establishing the role of dopamine as the teaching signal proposed in reinforcement learning theories.

Comparatively fewer experiments have used combinations of aversive stimuli and rewarding stimuli to characterize the dopamine response. *Fiorillo (2013)* proposed that reward and aversion are processed separately in the brain, based on the observation that dopamine neurons signaled information about reward but largely ignored aversive events (*Fiorillo, 2013*). This result contradicts some previous studies that showed consistent inhibition of dopamine neurons by aversive stimuli or the cues that predict them (*Matsumoto and Hikosaka, 2009*; *McCutcheon et al., 2012*; *Roitman et al., 2008*). Our results suggest that there are different modes of dopamine signaling: in one mode, dopamine neurons indeed integrate the information about reward and aversion and signal VPE. This is an ideal teaching signal for reinforcement learning to maximize future values. Further, we found that the response function to aversive stimuli was preserved across dopamine neurons, suggesting that each dopamine neuron has the potential to contribute a prediction error of aversiveness, as well as of reward (*Eshel et al., 2016*).

However, in a high reward context, dopamine neurons largely lose their ability to signal integrated VPEs. Our results in high reward contexts are consistent with those observed previously

(*Fiorillo, 2013*), and we also showed that similar results were obtained in previous recordings of optogenetically-identified dopamine neurons (*Cohen et al., 2012*; *Tian and Uchida, 2015*). This raises the possibility that one of the apparent differences observed between previous electrophysiological studies is due to different experimental parameters with respect to reward contexts. It should be noted that many physiological experiments tend to include highly rewarding training sessions in order to motivate animals. In natural environments in which wild animals forage, rewards might not be as abundant as in these experimental conditions. Our results indicate that dopamine neurons signal VPE with high fidelity in low reward contexts.

In examining the temporal dynamics of dopamine responses, we realized that on average, the peak of excitation for reward cues occurred earlier than the trough of inhibition for aversive cues. Interestingly, dopamine responses to the cue predicting both rewarding and aversive outcomes in our mixed prediction tasks first showed excitation and then inhibition, different from the flatter responses to nothing cues. This clear temporal difference raises the possibility that information about values from rewarding and aversive outcomes are not yet integrated in presynaptic neurons and arise from different sources of inputs to dopamine neurons. A recent electrophysiological recording study from monosynaptic inputs to dopamine neurons also suggested that different presynaptic neurons may convey values for rewarding versus aversive stimuli (*Tian et al., 2016*). Consistent with this idea, we did not observe a correlation between the magnitude of single dopamine neurons' responses to reward and aversiveness (*Figure 4C*), in contrast to correlations within reward-related responses (*Eshel et al., 2016*) and within aversiveness-related responses (*Figure 4A, B*). Of note, a previous study (*Eshel et al., 2016*) showed that, in high reward context, neurons that were highly responsive to unexpected rewards tended to also be highly responsive to aversive events: they showed greater levels of suppression below baseline. The results in the previous study are reminiscent of Fiorillo's study, which found that the inhibitory phase in biphasic responses of dopamine neurons did not encode negative values of aversive stimulus, but rather encode 'no reward' (*Fiorillo, 2013*). The results in the previous study (*Eshel et al., 2016*) could be consistent with the present results if the inhibition represents 'no reward' but not the negative value of aversive stimulus, and this no-reward response is correlated with other reward-related responses of dopamine neurons.

## Reward-context dependent representation in dopamine neurons

Our results indicate that dopamine neurons represent aversive information in a reward context dependent manner. Our results are consistent with a previous study which proposed that dopamine neurons change their response patterns depending on reward context (*Kobayashi and Schultz, 2014*). The authors found that neutral stimuli excited dopamine neurons more strongly in high- compared to low-reward contexts (*Kobayashi and Schultz, 2014*). The present study extends this finding to aversive stimuli. Short latency excitations of dopamine neurons have been observed in various experiments and have been attributed to generalization (*Kobayashi and Schultz, 2014*; *Mirenowicz and Schultz, 1996*), stimulus intensities (*Fiorillo et al., 2013*), motivational salience (*Bromberg-Martin et al., 2010*; *Matsumoto and Hikosaka, 2009*), trial starts (*Bromberg-Martin et al., 2010*) or stimulus detection (*Nomoto et al., 2010*). Our data do not distinguish these possibilities and the short latency excitation in the high reward context is likely to comprise a combination of these. Importantly, however, our data in the high reward context showed that the short-latency excitations compromised the monotonic value coding of dopamine neurons, and the difference between responses to air puff-predicting CS and nothing-predicting CS was diminished. This means that dopamine neurons did not simply add a constant amount of spikes (the same amount of excitation) on top of the monotonic value coding. Thus, our observations suggest that the combination of these factors and/or additional factors distorted normal value coding in dopamine neurons in high reward context.

In our experiments, high- and low-reward contexts differed with respect to the probability of rewarded trials. This suggests that dopamine responses to aversion depend on the frequency of reward, which may in turn change the animal's state. It remains to be examined how the frequency of rewards changes dopamine responses and whether dopamine responses could be modulated by other manipulations of the environment such as the amount of reward or the strength or frequency of aversive events.

In addition to overall reward contexts, we found that the inhibitory responses of dopamine neurons changed on a finer time-scale; the inhibition of dopamine neurons by air puff-predicting cues was correlated with trial-by-trial variability of aversion-related behaviors in low reward contexts. A similar correlation was observed between excitation of dopamine neurons by reward-predicting cues and reward-related behaviors. These results suggest that dopamine neurons track the predictions of values (reward and aversiveness) which may reflect animals' states over various time-scales.

A previous study also examined dopamine responses to aversive stimuli in relation to behaviors. Using cyclic voltammetry, the authors showed that, in response to electrical shock-predicting cues, the dopamine concentration in the ventral striatum increased when the rats exhibited an active avoidance behavior while it decreased when the rats showed freezing behavior (*Oleson et al., 2012*). It is therefore proposed that dopamine responses depend on whether the animal exhibits active avoidance or passive reaction (*Oleson et al., 2012*; *Wenzel et al., 2015*). In the present study, we found that the degree of inhibition, not excitation, of dopamine neurons in response to the air puff-predicting CS was positively correlated with anticipatory eye-blinking behaviors (*Figure 6D*). According to the above idea (*Oleson et al., 2012*; *Wenzel et al., 2015*), the anticipatory eye-blinking that we observed may be categorized as a passive avoidance behavior, which could be the reason as to why we observed inhibition, but not excitation of dopamine neurons correlated with anticipatory eye-blinking behaviors.

### Diversity of dopamine neurons

Whereas dopamine neurons displayed a relatively uniform response function to aversion in low reward contexts, we observed diverse responses in high reward contexts, including some inhibitory and some excitatory responses to aversive events. What caused diverse responses to aversion in high reward contexts? Increasing evidence supports the diversity of dopamine neurons depending on the location of the cell body and projection targets (*Lammel et al., 2014*; *Roeper, 2013*). For example, it was reported that neurons in the lateral SNc signal salience (*Lerner et al., 2015*; *Matsumoto and Hikosaka, 2009*) or 'stable value' as opposed to 'flexible value' in the medial SNc (*Kim et al., 2015*). Another study showed that dopamine neurons in the ventromedial VTA exhibited excitation to an aversive stimulus (*Brischoux et al., 2009*). Previous studies showed that responses to aversive stimuli are diverse across dopamine neurons with different projection targets (*Lammel et al., 2011*; *Lerner et al., 2015*). Although the majority of dopamine neurons in the lateral VTA, our main recording site, project to the ventral and anterior dorsal striatum (*Lammel et al., 2008*; *Menegas et al., 2015*), our study did not distinguish the exact projection targets of dopamine neurons. It remains to be determined which subpopulations of dopamine neurons switch signaling modes depending on low versus high reward contexts.

Our results demonstrated the importance of considering global contexts and behaviors and of unambiguously identifying dopamine neuron. It remains to be examined in future studies how reward frequency changes both the animal's state and dopamine responses to punishment, and how these changes relate to our normal and abnormal behaviors. Further, there are complex temporal dynamics and diversity of dopamine activities. Considering these factors together is challenging but represents a firm step towards fully understanding the nature and function of dopamine signals.

## Materials and methods

### Animals

We used 15 adult male mice heterozygous for Cre recombinase under control of the <Slc6a3> gene that encodes the dopamine transporter (DAT) (B6.SJL-*Slc6a3*$^{tm1.1(cre)Bkmn}$/J, Jackson Laboratory; RRID:IMSR_JAX:006660) (*Bäckman et al., 2006*). All mice were backcrossed with C57/BL6 mice. Eleven out of 15 mice were further crossed with tdTomato-reporter mice (*Gt(ROSA)26Sor*$^{tm9(CAG-tdTomato)Hze}$, Jackson Laboratory) to express tdTomato in dopamine neurons. Electrophysiological data were collected from 14 mice, and video data were from 8 mice. Animals were singly housed on a 12 hr dark/12 hr light cycle (dark from 06:00 to 18:00) and each performed the conditioning task at the same time of day, between 08:00 and 16:00. All procedures were approved by Harvard University Institutional Animal Care and Use Committee.

## Surgery and viral injections

Total 1 μl of adeno-associated virus (AAV), serotype 5, carrying an inverted ChR2 (H134R)-EYFP flanked by double *loxP* sites (*Atasoy et al., 2008*) [AAV5-DIO-ChR2-EYFP (*Tsai et al., 2009*)] was injected stereotactically into the VTA (3.1 mm posterior to bregma, 0.5 mm lateral, 3.9 mm deep from dura and 3.5 mm posterior to bregma, 0.5 mm lateral, 4.2 mm deep from dura). We previously showed that expression of this virus in dopamine neurons is highly selective and efficient (*Cohen et al., 2012*).

After > 1 week from virus injection, a custom-made metal plate (a head plate) was implanted. A microdrive containing electrodes and an optical fiber was implanted in the VTA stereotactically in the same surgery. All the surgeries were performed under aseptic conditions under isoflurane inhalation anesthesia (1–2% at 0.8–1.0 L min$^{-1}$). The animals were given analgesics (ketoprofen, 1.3 mg kg$^{-1}$ intraperitoneal, and buprenorphine, 0.1 mg kg$^{-1}$ intraperitoneal) postoperatively.

## Behavioral task

After >1 week of recovery, mice were water-deprived in their home cage. Animals were habituated for 1–2 days with the head restrained by a head plate before training on the task. Odors were delivered with a custom-designed olfactometer (*Uchida and Mainen, 2003*). Each odor was dissolved in mineral oil at 1:10 dilution. 30 μl of diluted odor was placed inside a filter-paper housing (Thomas Scientific, Swedesboro, NJ). Odors were selected pseudorandomly from isoamyl acetate, eugenol, 1-hexanol, citral, and 4-heptanone for each animal. Odorized air was further diluted with filtered air by 1:8 to produce a 900 ml min$^{-1}$ total flow rate.

We delivered an odor for 1 s, followed by 1 s of delay and an outcome. Trials were pseudorandomly interleaved. In the mixed prediction task, during initial training period (1–3 days), an odor (not used for Odors A–D) preceded a drop of water (4 μl) and Odor B preceded no outcome. After the initial training, Odors A–D were paired with water with 25% probability, no outcome (100% nothing), air puff with 75% probability, and water with 25% probability and air puff with remaining 75% probability. In high and low reward probability tasks, 2 odors (not used for Odors A–C) preceded a drop of water and no outcome, respectively, during initial training period. Later, Odor A was paired with water with 90% probability in the high reward probability task, and 20% probability in the low reward probability task. The probabilities of no outcome in Odor B trials and air puff in Odor C trials were 100% and 90%, respectively, in both reward probability tasks. In high reward probability task 2, 2 odors (Odor A and Odor B) preceded a drop of water and no outcome, respectively, during initial training period. Later, Odor A was paired with water with 90% probability and Odor C was paired with air puff with 80% probability. Air puff was delivered to the animal's right eye. The strength of air puff was enough to cause anticipated eye-blinking behavior. In order to control any sounds caused by air puff accumulation, the air was accumulated at the offset of odor delivery in all the trial types and released at 2.3 s after the offset of odor delivery outside of a hemi-soundproof behavioral box except for air puff trials. Licks were detected by breaks of an infrared beam placed in front of the water tube.

To quantify eye-blinking behavior trial-by-trial, animal's face including right eye was recorded by a CCD camera (Point Grey). The sampling rate was 60 Hz. To monitor animal's eye under dark conditions, we put infrared light sources inside the behavior box. To synchronize the video frames with event time stamps for further analysis, the infrared light source was turned off briefly (<25 ms) 2 s after the onset of US.

We also added no-odor trials (4% of the trials in the mixed prediction task, 12% in high and low reward probability tasks, and 13% in the high reward probability task 2) in which either water reward or air puff was presented unpredictably. Inter-trial intervals (ITIs) were drawn from an exponential distribution, resulting in a flat ITI hazard function truncated at 15 s such that expectation about the start of the next trial did not increase over time. Data in the mixed prediction task were obtained from 63 sessions (19–25 sessions per animal, 21 ± 3 sessions; mean ± S.D., $n$ = 3 mice); data in the high reward probability task were obtained from 38 sessions (2–19 sessions per animal, 10 ± 8 sessions, $n$ = 4 mice); data in the low reward probability task were obtained from 20 sessions (2–10 sessions per animal, 7 ± 4 sessions, $n$ = 3 mice); data in the high reward probability task 2 were obtained from 39 sessions (1–22 sessions per animal, 10 ± 9 sessions, $n$ = 4 mice). The animals performed between 208 and 476 trials per day (371 ± 81 trials; mean ± S.D.) in the mixed prediction

task, 272 trials per day in both high and low reward probability tasks, and between 182 and 454 trials per day (322 ± 42 trials; mean ± S.D.) in the high reward probability task 2.

## Electrophysiology

We recorded extracellularly from multiple neurons simultaneously using a custom-built 200-µm-fiber-optic-coupled screw-driven microdrive with six or eight implanted tetrodes (four wires wound together). Tetrodes were glued to the fiber optic (Thorlabs) with epoxy (Devcon). The ends of the tetrodes were 350–500 µm from the end of the fiber optic. Neural signals and time stamps for behavior were recorded using a DigiLynx recording system (Neuralynx). Broadband signals from each wire filtered between 0.1 and 9000 Hz were recorded continuously at 32 kHz. To extract the timing of spikes, signals were band-pass-filtered between 300 and 6000 Hz. Spikes were sorted off-line using MClust-3.5 software (David Redish). At the end of each session, the fiber and tetrodes were lowered by 20–80 µm to record new neurons. Sessions of recordings were continued until the tetrodes reached the bottom of the brain where no units were recorded and large fluctuations of voltage traces were recorded from tetrodes. After the completion of the recording sessions, tetrodes were moved up to the depth where units were recorded or the depth where light-identified dopamine neurons were recorded to ensure that the following electrolytic lesions were in the brain.

To verify that our recordings targeted dopamine neurons, we used ChR2 to observe stimulation-locked spikes (*Cohen et al., 2012*). The optical fiber was coupled with a diode-pumped solid-state laser with analogue amplitude modulation (Laserglow Technologies). Before and after each behavioral session, we delivered trains of 10 light pulses, each 5-ms long, at 1, 2, 5, 10, 20 and 50 Hz at 473 nm at 5–20 mW mm$^{-2}$. Spike shape was measured using a broadband signal (0.1–9000 Hz) sampled at 30 kHz. This ensured that particular features of the spike waveform were not missed.

We used two criteria to include a neuron in our data set. First, the neuron must have been well isolated [L-ratio < 0.05 (*Schmitzer-Torbert and Redish, 2004*), except for two units with L-ratio = 0.055 and 0.057]. Second, the neuron must have been recorded in or between the sessions when dopamine neurons were identified on the same tetrode to ensure that all neurons came from VTA. Recording sites were further verified histologically with electrolytic lesions using 5–20 s of 30 µA direct current and from the optical fiber track. Recording sites of individual dopamine neurons were reconstructed on the Franklin and Paxinos brain atlas (*Franklin and Paxinos, 2008*). The depths were estimated from the lesion site in each animal.

## Data analysis

To measure firing rates, peristimulus time histograms (PSTHs) were constructed using 1-ms bins. To calculate spike density functions, PSTHs were smoothed using a box filter (100 ms duration, t ± 50 ms). Average firing rates of responses to conditioned stimulus (CS) were calculated using a time window of 0–1000 or 200–1000 ms after odor onset. To obtain responses to the unconditioned stimulus (US), we used a time window of 0–600 ms after the onset of US except that responses to air puff omission and nothing (no outcome) were calculated using a time window 0–1000 ms. Slightly different window sizes were also tested and gave qualitatively the same results. The baseline firing rates were obtained based on the activity in a time window during inter-trial-interval immediately preceding odor onset (-1000 to 0 ms before odor onset). The baseline firing rates were computed by using data from all trial types.

We calculated the area under the receiver-operating characteristic (auROC) value of each neuron using the trial-by-trial responses to CS, unpredicted and predicted outcomes in time windows previously described.

The area of the right eye region was calculated as follows (see also *Figure 6—figure supplement 1*): (1) Eye threshold: Since in our recording settings, most of the face background area was saturated (close to 255 pixel intensity), a threshold around 234 pixel intensity was used to separate eye area from the background. Pixels with intensity smaller than the threshold were set to 1 (white) and others were set to 0 (black). (2) Remove dark patches outside of the eye: To remove the occasional dark patches outside of eye area in the raw image (e.g., top panel in *Figure 6—figure supplement 1A*), connected areas smaller than 500 pixels were deleted. (3) Smooth the eye patch: We performed morphological opening to remove spiky edges. Then we filled all black spots on the binary image (e.g., in *Figure 6—figure supplement 1C*) smaller than 500 pixels to remove the bright spots

inside of the eye area due to reflection. (4) Compute eye area: We found the largest connected regions on the binary image in *Figure 6—figure supplement 1D* and computed the area of this region in pixels and also computed the eccentricity by fitting the area to an eclipse (MATLAB region-props function). These codes for extracting eye areas from video files are available at https://github.com/hide-matsumoto/prog_hide_matsumoto_2016.

To analyze the eye-blinking behavior trial-by-trial, we synchronized video frames with Neuralynx timestamps as follows: (1) Detect frames when infrared light was off. When infrared light source was briefly turned off (<25 ms), average pixel intensity of the frame steeply decreased. Thus, when the average pixel intensity of each frame in the session was plotted over time, the light-off frames were detected as troughs of the value (*Figure 6—figure supplement 1E*). (2) We then matched these frames that have troughs of average pixel intensity with time stamps of infrared light source-off saved in Neuralynx. (3) Interpolate time of other frames: The timing of a video frame was interpolated using the time stamps of the two closest light-off frames. The eye areas extracted from video frames were further analyzed using event time stamps saved in Neuralynx.

To compare eye areas across sessions, the computed eye areas were normalized by the maximum eye area (99th percentile of all the eye areas) in every session. Trials were categorized into two groups, blink and no-blink trials, using the criteria that the averaged eye area during delay period in each air puff trial was larger (no-blink trials) or smaller (blink trials) than 0.5.

To check the percentage of trials that the animals showed anticipatory licking during the delay period (1–2 s from odor onset), trials were categorized into two groups, lick and no-lick trials, using the criteria that the lick rate during the delay period was larger (lick trials) or smaller (no-lick trials) than 3. The threshold (3 licks $s^{-1}$) was determined by comparing the distributions of the lick rates during the delay period in rewarded trials and those in nothing trials (*Figure 6—figure supplement 4A*).

For each statistical analysis provided in the manuscript, the Kolmogorov–Smirnov normality test was first performed on the data to determine whether parametric or non-parametric tests were required. Data were analyzed in MATLAB (MathWorks) and were shown as mean ± S.E.M., unless otherwise stated. For unpaired $t$ test, the equality of variance between two groups was first validated statistically. For paired and unpaired comparisons, two-sided tests were used. Bonferroni correction was applied for significance tests with multiple comparisons. To test monotonicity of CS responses, we chose neurons showing that (1) their CS responses were significantly modulated by odors (examined by one-way ANOVA), (2) the response to reward-predicting CS was significantly larger than that to air puff-predicting CS ($p<0.05$, unpaired $t$ test), and (3) the averaged response to CS predicting nothing (or CS predicting both reward and air puff) was intermediate between that to reward-predicting CS and that to air puff-predicting CS. Sample sizes in this study were based on previous literature in the field (*Cohen et al., 2012*; *Eshel et al., 2015*, *2016*; *Tian and Uchida, 2015*) and were not pre-determined by a sample size calculation. Randomization and blinding were not employed.

## Immunohistochemistry

After recording, mice were transcardially perfused with saline and then with 4% paraformaldehyde under anesthesia. Brains were cut in 100 μm coronal sections. Brain sections from DAT-cre mice were immunostained with antibodies to tyrosine hydroxylase (AB152, 1:400, Millipore; RRID:AB_390204) and secondary antibodies labeled with Alexa594 (1:200, Invitrogen) to visualize dopamine neurons. Sections were further stained with 4′,6-diamidino-2-phenylindole (DAPI, Vector Laboratories) to visualize nuclei. Recording sites were further verified to be amid EYFP- and tdTomato-positive or tyrosine hydroxylase-positive neurons in VTA.

## Acknowledgements

We thank H Kim for eye monitoring system, C Starkweather, N Eshel and other members of the Uchida lab for discussions and C Dulac for sharing resources. This work was supported by a fellowship from Japan Society for the Promotion of Science (HM), the Uehara Memorial Foundation (HM), the Sackler Scholar Programme in Psychobiology (JT) and NIH grants R01MH095953 (NU), R01MH101207 (NU), and R01MH110404 (NU).

# Additional information

## Competing interests
NU: Reviewing editor, *eLife*. The other authors declare that no competing interests exist.

## Funding

| Funder | Grant reference number | Author |
|---|---|---|
| Japan Society for the Promotion of Science | JSPS postdoctoral fellowship for research abroad | Hideyuki Matsumoto |
| Uehara Memorial Foundation | The Uehara Memorial Foundation postdoctoral fellowship | Hideyuki Matsumoto |
| Sackler Scholar Programme in Psychology | Predoctoral fellowship | Ju Tian |
| National Institute of Mental Health | R01MH095953 | Naoshige Uchida |
| National Institute of Mental Health | R01MH101207 | Naoshige Uchida |
| National Institute of Mental Health | R01MH110404 | Naoshige Uchida |

The funders had no role in study design, data collection and interpretation, or the decision to submit the work for publication.

## Author contributions
HM, Performed video analysis of eye blinking behaviors, Conception and design, Acquisition of data, Analysis and interpretation of data, Drafting or revising the article; JT, Performed video analysis of eye blinking behaviors; NU, MW-U, Conception and design, Analysis and interpretation of data, Drafting or revising the article

## Author ORCIDs
Naoshige Uchida, http://orcid.org/0000-0002-5755-9409
Mitsuko Watabe-Uchida, http://orcid.org/0000-0001-7864-754X

## Ethics
Animal experimentation: This study was performed in strict accordance with the recommendations in the Guide for the Care and Use of Laboratory Animals of the National Institutes of Health. All of the animals were handled according to approved Harvard animal care and use committee (IACUC) protocols (#26-03) of Harvard University. All surgery was performed under isofluorane anesthesia, and every effort was made to minimize suffering.

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
