## [Decision Letter]

Thank you for submitting your article "Midbrain dopamine neurons signal aversion in a reward-context-dependent manner" for consideration by *eLife*. Your article has been reviewed by three peer reviewers, and the evaluation has been overseen by a Reviewing Editor and Timothy Behrens as the Senior Editor. The reviewers have opted to remain anonymous.

The reviewers have discussed the reviews with one another and the Reviewing Editor has drafted this decision to help you prepare a revised submission.

The reviewers appreciated the new data evaluating the relationship between aversive and appetitive coding in dopamine neurons of the VTA in this study, and also the effort in doing these analyses with carefully identified dopamine. However, the overall opinion was that the authors have to more carefully analyze their existing data, and to ensure the data clearly support the conclusions they make. Below is a summary of the 3 main types of concern that should be addressed:

1) A general comment brought up by all reviewers is that the authors should analyze the phasic neuronal responses with better temporal specificity, i.e., not just averaging over one larger time bin. The reviewers felt that it is not justified to ignore temporal dynamics in the aversive CS response, and that the conclusions (for example regarding stimulus generalization) will be changed/tempered if this is done.

2) The reviewers felt that the relationship between the behavior and the neural activity was not being fully fleshed out, and so new analyses are important to substantiate some of the claims. Namely relationship between CS and US in different predicted and unpredicted conditions, meaning of control tasks, and analyses of anticipatory licking, etc.

3) Another concern was the small n for many conclusions and the need for some additional statistical analyses. Although new experiments may be beyond the scope of this review, some conclusions may be biased by a low n (no significance in some cases vs. significance in other cases). Also, ANOVA and multiple comparison correction may be needed in some cases.

Also below are the detailed points of each reviewer to help in your revised submission.

*Reviewer #1:*

Matsumoto and colleagues describe the responses of identified dopamine (DA) neurons recorded from the lateral VTA of mice during multiple variations of an odor discrimination task. The methodology, including the DA neuron identification and recording, is similar to prior work from this group, and reflects a careful and sophisticated approach towards investigating the responses of individual identified neurons to outcomes and the cues that predict them. The authors' aim is to investigate the encoding by DA neurons of aversive stimuli, given the mixed findings in the literature. The approach taken here is to compare the neuronal responses to cues in a discrimination procedure in which reward-predictive and airpuff-predictive cues, as well as cues predicting both, or nothing, are presented pseudorandomly. The authors find that the encoding of aversive-predicting cues varies with the relative probability of other cues being reinforced by reward, such that DA response to airpuff-paired cues are decreases in spiking in low density reward sessions, and DA responses to airpuff-paired cues are increases in spiking in high density reward sessions (although decreases are also clearly visible in the neural response – see below). These are interesting findings. The results do add substantially to our understanding of DA neural responses by providing careful parametric comparisons in, very critically, *identified* DA neurons, using opto-tagging. Importantly, the notion that overall reward density may alter responses to cues – reward paired, or not – was shown previously by Schultz and colleagues, and this is acknowledged by the authors. The present data add to our understanding of the regulation of DA neuron responses to reward- and punishment-paired cues and will be of strong interest to scientists studying dopamine systems, as well as more generally, reward, addiction, and learning. The paper would benefit from more careful consideration of the specific attributes and limitations of these studies in forming their interpretations.

In addition, an explanation of how the differing signals in response to the negative cue in high and low reward contexts might be expected to impact current or future behavior differentially is missing, given that the behavioral data presented indicate that anticipatory eye blinking is similar in both contexts – however their conclusion on this point may be a function of their analysis approach. Specifics follow.

1) The authors conclude that "In low reward contexts, dopamine neurons were inhibited by aversive events more strongly than by neutral events, tracking aversiveness of stimuli on a trial-by-trial basis […] However, in high reward contexts, dopamine neurons exhibited early excitations, which undermined their ability to produce negative responses for aversive events." While their data as currently analyzed support this statement, it begs the question of what these different "modes" mean for learning and behavior since they presumably would communicate different information to downstream structures. The authors show that mice tend to show anticipatory blinking for airpuff in response to the cue whether it is a low or high reward context, but the relationship between the suppression of firing in response to the airpuff-predictive cue only holds for the low reward context, because consistent suppression in the DA neuron response is not seen in response to this cue in the high reward context. Yet this does not take the biphasic nature of the neural response to the aversive stimulus in the high reward context into account. In light of this (as well as the comment that follows), the pupillary/blink response appears to occur on a timescale parallel to that of the late component of the neural response, which in the case of the aversive cue, is an inhibition. Thus, it appears to me that there may indeed by a correlation with behavior and the neural signal. Can the authors investigate this issue more deeply?

2) The authors further state that: "[…]dopamine neurons signaled VPEs by faithfully combining information about both appetitive and aversive events into a common currency of value." Their evidence for this is that the average firing rate of DA neurons following a cue that predicts either an airpuff or reward is between that of a reward-predicting and an airpuff-predicting cue – however, this conclusion may depend overmuch on the averaging of the signal over the first second of cue presentation. Examination of the population histograms clearly show that this signal is dynamic, and that averaging the response over the full first second of cue presentation may obscure a multiphasic response to the aversive cues – there looks to be an excitation followed by an inhibition in the high reward context. Again, this is a response pattern described by Schultz and colleagues, who suggests that the early response reflects attributes such as stimulus generalization and intensity, and it is the later response that reflects the prediction error/value signal. This interpretation should receive deeper treatment in the current manuscript. Maybe the averaging across the full second is the most direct statistical approach, but the biphasic nature of the signal should be acknowledged.

3) It has been proposed that excitatory responses to cues that do not predict reward in multi-cue procedures (wherein some cues do predict reward) reflect stimulus generalization, among other factors. This seems to be one possible explanation for the finding that increases in response to the airpuff cue are more often seen in a high reward context than a low reward context. In addition, the authors data further support this notion since the cue that predicts no reward also shows an initial excitation in the high reward, but not low reward, context. Have the authors thought to use cues of differing modalities to reduce the sensory similarity? If not, the authors should consider that their data do not contradict a generalization and/or salience notion, although they appear to say they do.

4) The authors conclude that "dopamine neurons do not encode aversion in a conventional classical conditioning task". Given the clear evidence provided by the authors that reward context impacts the response of VTA dopamine neurons provided by the authors, the authors have not provided an appropriate test of this hypothesis. To conclude this, the authors would need to conduct their recordings in a procedure without interleaved rewarded trials.

5) I am curious what the potential differences in the responses to the conditioned stimuli and the unconditioned stimuli for a given neuron might be. Indeed, it would be useful if the authors reported the extent to which the encoding observed during the cue is similar to encoding during the outcome, i.e., if a neuron is inhibited by an airpuff-predicting cue, is it also inhibited by unexpected air puff. This is especially interesting given the points above, that the high reward context increase in firing at the outset of the odor cues seems parsimoniously explained as generalization, while the neural responses to the respective outcomes could still be increases and decreases for sucrose and airpuffs, respectively. In other words, do DA neuron responses to presentation of airpuff itself change in different reward contexts, or is this strictly a feature of the conditioned stimuli.

*Reviewer #2:*

In this manuscript, Matsumoto et al. investigated the response of midbrain dopaminergic neurons to an aversive stimulus in a variety of contexts. Although dopamine is thought to be important for learning from both reward and punishment, there has been disagreement about how dopaminergic neurons integrate aversive and rewarding stimuli. The authors show that dopaminergic activity is inhibited by an air puff and an air puff-predicting odor in low reward contexts. The air puff response is modulated by expectation and individual dopaminergic neurons are able to integrate the value of reward and punishment. Finally, they show that manipulating reward context affects dopaminergic responses to punishment, such that in high reward contexts there is actually an excitatory response to punishments. The latter condition appears to replicate various previous studies. The data provides a valuable contribution to the debate over how dopaminergic neurons respond to aversive stimuli. The use of optogenetic tagging allows for compelling identification of dopaminergic neurons and the authors tested several task structures in order to reconcile discrepancies in the field.

There are several ways in which the analysis and presentation should be improved:

1) In Figure 2 the authors conclude that in a high reward context, dopaminergic neurons respond to aversive stimuli with short-latency excitation "masked inhibitory responses" such that there is no longer a significant difference between responses to odor B and odor C. The importance of reward context is a major conclusion of the paper but its complete disruption of aversive signaling is confusing (and not entirely convincing) as presented. Why not analyze the different temporal phases of the response separately? It appears that if the responses were compared during the late inhibitory phase of the response (evident in Figure 2), then there would indeed be a difference in the response to odor B and odor C in both the high reward and low reward contexts. In other words, the simplest explanation of the effect of "high reward context" is to create a short-latency excitatory response to the aversive stimulus. The authors argue this response is different from the short-latency 'generalization' response described by Schultz and colleagues, but not clear what that means. Specifically, in the Discussion it is noted that a high reward context "compromised the monotonic value coding of dopamine neurons," but this was not clearly explained or analyzed. Additionally, the argument that the excitatory response to punishment in a high reward context is not due to generalization is unclear. The authors write that "dopamine neurons did not simply add the same amount of excitation on top of the monotonic value coding." In fact, in Figure 2, it seems like there is an excitatory "generalization" response that is being added to the pause in firing to the CS-.

2) Although there is a clear increase in the amplitude of the response to air puff when it is unpredicted, there is large variability in the response to an omitted air puff. Although on average there is a very small increase in firing rate, a very large number of neurons decrease their activity in response to air puff omission (Figure 3). The authors should discuss the heterogeneity of this response, which appears to complicate their conclusions.

3) It seems surprising that there is no correlation between the response to unpredicted air puff and unpredicted water delivery (Figure 5). In Eshel et al. (2016) there is a correlation between predicted air puff and unpredicted water responses as well as between the air puff cue and the unpredicted water responses. So why wouldn't there be a correlation between unpredicted air puff and reward? Doesn't this mean that predicted and unpredicted air puff responses are themselves uncorrelated, which would contradict the earlier panels in Figure 5?

4) The protocol for the classical conditioning experiment conducted for Figure 2—figure supplement 1 is unclear. It appears that the only difference from the high reward task is the probability of punishment following odor C which is 80% instead of 90%, yet the excitatory component of the response to the punishment-predicting CS appears considerably larger than in Figure 2.

5) Not clear exactly what the eye blink response analysis achieves in the paper, nor why eye blinks were analyzed but not anticipatory licking.

6) There is no reported method for the correction for multiple comparisons in multiple instances when more than 2 groups are being analyzed in the same panel. Additionally, a two-way ANOVA should be used to analyze the effect of reward probability and anticipatory blinking in Figure 4.

*Reviewer #3:*

In this manuscript Matsumoto and colleagues present a series of experiments that address the issue of how dopamine neurons respond to aversive events. They make a number of important and interesting observations. On the whole I thought the manuscript was well-written. I have a few concerns that I list below:

1) I feel that the Introduction would benefit from some elaboration in a few places.

Introduction, fourth paragraph: should make it clear that Oleson et al. (2012) measured dopamine release using FSCV.

Introduction, fifth paragraph: I am not clear how Roitman et al. (2008) resolved the issue and this was published several years earlier, so the wording is confusing in my view.

Introduction, fifth paragraph: should make it clear that Lerner et al. (2015) measured calcium activity, not action potential firing.

Introduction, last paragraph: this paragraph alludes to the issue of neurochemical identity in a way that I think should be more explicit since it is of particular relevance to this study. For example, Ungless et al. (2004) showed that some VTA neurons that are excited by aversive events are in fact not dopaminergic. It is noted, for example by Schultz (2016) in a recent review, that Matsumoto & Hikosaka (2009) recorded many neurons in areas where dopamine neurons were rare. This more strongly justifies the use of the opto-tagging approach that the authors have taken.

2) One general concern I had was the low numbers of mice in several of the experimental groups (e.g., Figure 1, n=3).

3) How do the authors know that the mice can tell the difference between the low and high reward situations?

4) In the subsection “Integration of aversiveness and reward in dopamine neurons”, first paragraph: add the caveat that activation of meso-cortical dopamine neurons and dorsal raphe dopamine neurons can induce conditioned-place aversion (Lammel et al. 2014 & Matthews et al. 2016).

4) In the subsection “Diversity of dopamine neurons”: note also that Brischoux et al. (2009) reported that ventromedial dopamine neurons exhibited aversive activations.

5) I found some of the red/purple colour schemes difficult to tell apart. I would suggest some different colour schemes (particularly for Figure 1).

[Editors' note: further revisions were requested prior to acceptance, as described below.]

Thank you for resubmitting your work entitled "Midbrain dopamine neurons signal aversion in a reward-context-dependent manner" for further consideration at *eLife*. Your revised article has been favorably evaluated by Timothy Behrens (Senior editor), a Reviewing editor, and two reviewers.

The manuscript has been improved but there are some remaining issues regarding its readability that need to be addressed before acceptance, as outlined below by Reviewer 2. After consultation between the reviewers, both thought that increased readability and explanation of why each experiment was performed would greatly enhance the clarity and impact of the study.

*Reviewer #2:*

I felt that the reviewers were responsive to our comments and that this study provides a valuable contribution to the discussion regarding the integration of rewarding and aversive cues in dopamine neurons.

My only remaining comment is that I find the text in the Results and Discussion a bit difficult to follow. There are a number of experiments/analyses, and as it currently stands, the motivation for each experiment/analysis, as well as the relevant conclusions, is not always completely transparent. Therefore, the authors could consider editing the Results/Discussion for improved readability.

*Reviewer #3:*

This new version addresses the previous concerns that I raised.

---

## [Author Response]

*The reviewers appreciated the new data evaluating the relationship between aversive and appetitive coding in dopamine neurons of the VTA in this study, and also the effort in doing these analyses with carefully identified dopamine. However, the overall opinion was that the authors have to more carefully analyze their existing data, and to ensure the data clearly support the conclusions they make. Below is a summary of the 3 main types of concern that should be addressed:*

We appreciate all of the reviewers for their constructive criticism and suggestions. We have performed additional analyses and modified our manuscript according to reviewer’s comments. We hope that our manuscript has improved.

*1) A general comment brought up by all reviewers is that the authors should analyze the phasic neuronal responses with better temporal specificity, i.e., not just averaging over one larger time bin. The reviewers felt that it is not justified to ignore temporal dynamics in the aversive CS response, and that the conclusions (for example regarding stimulus generalization) will be changed/tempered if this is done.*

We appreciate reviewers’ criticism on the choice of time windows. We are aware that, in previous studies, it has become a common practice to pick a time window somewhat arbitrarily, especially for the analysis of dopamine activities. Although these studies have shown that dopamine neurons often show multi-phasic responses and different information appears to be conveyed in different time windows, no study has shown that postsynaptic neurons are equipped with a special mechanism to separately read out information from these windows. We therefore chose to use a relatively large time window (0–1,000 ms) that covers the entire response period as this would be a more conservative approach, and used this method throughout the manuscript. This rationale is now explained in Results (subsection “Reward-context dependent representation of aversion in dopamine neurons”, fifth paragraph).

The above analyses used a relatively large time window that contains the entire response period (0–1,000 ms). Because dopamine responses in high reward contexts exhibited biphasic responses (early excitation followed by later inhibition), we further analyzed the data by separating these time windows into smaller bins. Because there is no known mechanism by which downstream neurons can read out these windows separately, analysis using a large window can be considered more conservative. However, previous studies have indicated that different information may be conveyed in these time windows (Schultz, 2016; Stauffer et al., 2016).

However, we agree that it is important to know what information is conveyed by different time windows, as Schultz recently proposed (Schultz, 2016, Nat Rev Neurosci). We therefore performed new analyses using the time window in which dopamine neurons are predominantly inhibited, i.e. the later response phase (200–1,000 ms). Analyses using the small time window generally confirmed that there is a similar trend. For instance, fewer neurons showed a significant difference between nothing-predictive- and air puff-predictive CSs in a high reward context. These results are now shown in Figure 4—figure supplement 2, and discussed in Results as follows:

“We obtained similar results even if we compared only later time bins (200–1,000 ms), excluding the early excitation phase (Figure 4—figure supplement 2). […] Neurons that showed a large excitation to the air puff CS were not necessarily the same group of neurons which showed excitation to the air puff itself, consistent with a previous study (Matsumoto and Hikosaka, 2009) (Figure 4—figure supplement 3).”

We have also performed analyses on the relation between trial-to-trial variability of dopamine activities and anticipatory blinking using the later time window (200–1,000 ms). This analysis is now shown in Figure 5—figure supplement 3 and discussed in Results as follows:

“The correlation between trial-by-trial dopamine activity and anticipatory blinking was even clearer if we consider reward contexts (Figure 5). […] The results do not change even when we only used a later window of dopamine CS responses, excluding the early excitation period (0–200 ms) (Figure 5—figure supplement 3).”

2) The reviewers felt that the relationship between the behavior and the neural activity was not being fully fleshed out, and so new analyses are important to substantiate some of the claims. Namely relationship between CS and US in different predicted and unpredicted conditions, meaning of control tasks, and analyses of anticipatory licking, etc.

We found that both reward context and behavioral outcome (blink) were important for dopamine neurons to show inhibition to mildly aversive cues but one was not sufficient. We rephrased this part both in Results and Discussion. We also added a new analysis in Figure 4—figure supplement 3 for the analysis of the relationship between CS and US, and Figure 5 and Figure 5—figure supplement 4 for the analysis of the relationship between anticipatory licking behavior and responses to reward CS. For more details of each analysis, please see our Response to each reviewer’s comment.

*3) Another concern was the small n for many conclusions and the need for some additional statistical analyses. Although new experiments may be beyond the scope of this review, some conclusions may be biased by a low n (no significance in some cases vs. significance in other cases). Also, ANOVA and multiple comparison correction may be needed in some cases.*

In this study, we used 14 mice for electrophysiological experiments, recorded from 176 neurons and identified 72 dopamine neurons in total. We agree that some of the task conditions contain relatively fewer number of neurons. However, for most of the analyses, we replicated the same results using two separate sets of experiments. To demonstrate the robustness of our basic findings, now we show data analyses using combined data in Figure 4—figure supplement 1 and Las well as side-by-side comparisons of comparable task conditions (Figure 4—figure supplement 1).

We performed one-way ANOVA for comparing the means of CS responses (1: reward-, nothing-, versus air puff-predicting CSs and 2: reward-, reward + air puff-, versus air puff-predicting CSs). We described the results in Results (subsection “Dopamine neurons integrate values of both valences, appetitive and aversive”, last two paragraphs). Also, Bonferroni correction was applied for significance tests with multiple comparisons. We added this sentence in Materials and methods (subsection “Data analysis”, last paragraph).

*Also below are the detailed points of each reviewer to help in your revised submission.*

*Reviewer #1:*

*1) The authors conclude that "In low reward contexts, dopamine neurons were inhibited by aversive events more strongly than by neutral events, tracking aversiveness of stimuli on a trial-by-trial basis […] However, in high reward contexts, dopamine neurons exhibited early excitations, which undermined their ability to produce negative responses for aversive events." While their data as currently analyzed support this statement, it begs the question of what these different "modes" mean for learning and behavior since they presumably would communicate different information to downstream structures. The authors show that mice tend to show anticipatory blinking for airpuff in response to the cue whether it is a low or high reward context, but the relationship between the suppression of firing in response to the airpuff-predictive cue only holds for the low reward context, because consistent suppression in the DA neuron response is not seen in response to this cue in the high reward context. Yet this does not take the biphasic nature of the neural response to the aversive stimulus in the high reward context into account. In light of this (as well as the comment that follows), the pupillary/blink response appears to occur on a timescale parallel to that of the late component of the neural response, which in the case of the aversive cue, is an inhibition. Thus, it appears to me that there may indeed by a correlation with behavior and the neural signal. Can the authors investigate this issue more deeply?*

Please also refer to our response to the editor’s point #1 regarding specific time windows. The question of “what these different modes mean for learning and behavior” is important. One possibility would be that dopamine directly controls or modifies some behaviors. In our task, however, temporal dynamics of dopamine activities and blinking are different: dopamine responses are tightly locked to salient events such as CS or US, while blinking occurs much later than dopamine responses. This appears to suggest that dopamine activities are not tightly time-locked with blinking behaviors. The suggested idea is that certain time window of inhibitory activities might cause blinking. We did not see this correlation between dopamine activities and blinking when we used a big time window. Now we have analyzed our data using only the later phase of responses in high reward contexts, and show that the results are very similar to those obtained using a big time window. We have now added this analysis in Figure 5—figure supplement 3.

We interpret these results to support the idea that the dopamine signal (value coding) is an abstract value representation rather than immediate trigger of actions. Blinking is a good indicator for mice predicting air puff but may not be perfectly correlated with the negative value of the air puff. Dopamine neurons may signal negative value but not action itself. As a result, depending on contexts, prediction of air puff may be correlated with dopamine activity (in low reward context) or not (in high reward context). We rephrased these parts for clarification in Results (e.g. subsection “Trial-to-trial variability dopamine responses to aversive stimuli”, third paragraph) and Discussion.

“The correlation between trial-by-trial dopamine activity and anticipatory blinking was even clearer if we consider reward contexts (Figure 5). [,,,] The results do not change even when we only used a later window of dopamine CS responses, excluding the early excitation period (0–200 ms) (Figure 5—figure supplement 3).”

*2) The authors further state that: "[…]dopamine neurons signaled VPEs by faithfully combining information about both appetitive and aversive events into a common currency of value." Their evidence for this is that the average firing rate of DA neurons following a cue that predicts either an airpuff or reward is between that of a reward-predicting and an airpuff-predicting cue – however, this conclusion may depend overmuch on the averaging of the signal over the first second of cue presentation. Examination of the population histograms clearly show that this signal is dynamic, and that averaging the response over the full first second of cue presentation may obscure a multiphasic response to the aversive cues – there looks to be an excitation followed by an inhibition in the high reward context. Again, this is a response pattern described by Schultz and colleagues, who suggests that the early response reflects attributes such as stimulus generalization and intensity, and it is the later response that reflects the prediction error/value signal. This interpretation should receive deeper treatment in the current manuscript. Maybe the averaging across the full second is the most direct statistical approach, but the biphasic nature of the signal should be acknowledged.*

Please also refer to the answer to the editor’s point #1. As the referee points out, it is important that previous studies have indicated that different information may be conveyed at different time windows. We now discuss the difference between the early and late phases of responses observed in the Schultz’s and Fiorillo’s studies in Introduction (sixth paragraph) and other places (e.g., in Results; subsection “Reward-context dependent representation of aversion in dopamine neurons”, second and fifth paragraphs).

As explained in our response to the editor’s point #1, we focused on a big time window because it remains unclear whether downstream neurons can read out information from specific, somewhat arbitrary response windows. In addition, our data indicates that defining a value- or prediction-error-specific time window is not very easy.

In our tasks, we did not observe a time window which solely signaled generalization and/or stimulus intensity regardless of the stimulus identity. For instance, Figure 1 shows that CS responses diverse in as quickly as 60 ms reflecting the predicted values of upcoming USs suggesting that our dopamine neurons can change their responses in a value dependent manner even in the early phase of the responses. The discrimination of odor identity in dopamine neurons in our task is fast, different from the previous tasks which used ambiguous cue on purpose (Nomoto et al., 2010, J Neurosci), and temporally overlaps with the early excitation during biphasic responses in high reward contexts.

Despite these difficulties in defining a time window, we have now performed analyses using only the late phase of the responses (200–1,000 ms). We did not observe value coding in this time window in response to air puff CSs in the high reward context, consistent with Fiorillo’s study (Fiorillo, 2013, Science). We added this result in Figure 4—figure supplement 2B. As for our mixed prediction tasks in Figure 1, if we split into two phases, the first phase includes many positive values of reward cues and the second phase includes many negative values of aversive cues, and so these patterns do not correspond to generalization and/or stimulus intensity and value, respectively. We think that this temporal shift of response peaks reflects the timing of responses encoding reward vs. aversion values with aversive information arriving slightly later, rather than the idea that the early period encodes generalization/intensity and the later period encodes values. We speculate that these shifted responses reflect the different sources of inputs signaling reward or aversion to dopamine neurons, which were not integrated yet in inputs. We have added these points in Discussion (page 17, lines 483-503).

“In examining the temporal dynamics of dopamine responses, we realized that on average, the peak of excitation for reward cues occurred earlier than the trough of inhibition for aversive cues. […] The results in the previous study (Eshel et al., 2016) could be consistent with the present results if the inhibition represents “no reward” but not the negative value of aversive stimulus, and this no-reward response is correlated with other reward-related responses of dopamine neurons.”

Because of the two limitations discussed above, we still prefer to report our main results, especially those pertaining to Figure 1, using a large time window.

*3) It has been proposed that excitatory responses to cues that do not predict reward in multi-cue procedures (wherein some cues do predict reward) reflect stimulus generalization, among other factors. This seems to be one possible explanation for the finding that increases in response to the airpuff cue are more often seen in a high reward context than a low reward context. In addition, the authors data further support this notion since the cue that predicts no reward also shows an initial excitation in the high reward, but not low reward, context. Have the authors thought to use cues of differing modalities to reduce the sensory similarity? If not, the authors should consider that their data do not contradict a generalization and/or salience notion, although they appear to say they do.*

Thank you very much for the comment and suggestion. We think that our data do not contradict with previous important findings about stimulus generalization etc. but rather expanded those findings. We tested how much these components disturb value coding of dopamine neurons. In addition to stimulus generalization, there are many other interpretations about the cause of early excitations. In each experiment, authors specifically designed to extract each factor from others. We think that the excitation that we observed in our tasks is caused by a combination of these factors, not only by stimulus generalization. The proposed experiment is an important experiment to understand which component is most affected by reward contexts but does not affect our statement that these components disturb value coding in dopamine neurons in high reward contexts. We revised our discussion on this point in the Discussion to be clearer (e.g. page 18, lines 507-524).

*4) The authors conclude that "dopamine neurons do not encode aversion in a conventional classical conditioning task". Given the clear evidence provided by the authors that reward context impacts the response of VTA dopamine neurons provided by the authors, the authors have not provided an appropriate test of this hypothesis. To conclude this, the authors would need to conduct their recordings in a procedure without interleaved rewarded trials.*

We apologize that this figure title was very misleading. We fixed this title. Thank you for letting us know.

*5) I am curious what the potential differences in the responses to the conditioned stimuli and the unconditioned stimuli for a given neuron might be. Indeed, it would be useful if the authors reported the extent to which the encoding observed during the cue is similar to encoding during the outcome, i.e., if a neuron is inhibited by an airpuff-predicting cue, is it also inhibited by unexpected air puff. This is especially interesting given the points above, that the high reward context increase in firing at the outset of the odor cues seems parsimoniously explained as generalization, while the neural responses to the respective outcomes could still be increases and decreases for sucrose and airpuffs, respectively. In other words, do DA neuron responses to presentation of airpuff itself change in different reward contexts, or is this strictly a feature of the conditioned stimuli.*

Thank you for the helpful comment. We observed similar phenomena in the responses to the unconditioned stimulus, i.e., air puff itself. In the low reward context, the response to air puff was mostly negative whereas the response was variable in the high reward context. Although we see a significant difference between responses to unpredicted air puff in high- and low-reward contexts using the neurons in the current data set (p = 0.001, n = 34 and n = 38, respectively, unpaired *t*-test), we would like to examine these phenomena more carefully because individual variability is big across neurons.

At the single neuron level, neurons which showed a large excitation to air puff CS were not necessarily the same group of neurons which showed excitation to air puff itself, consistent with a previous study (Matsumoto and Hikosaka, 2009, Nature). We now report this result in Figure 4—figure supplement 3 and in Results (page 12, lines 348-351).

“Neurons that showed a large excitation to the air puff CS were not necessarily the same group of neurons which showed excitation to the air puff itself, consistent with a previous study (Matsumoto and Hikosaka, 2009) (Figure 4—figure supplement 3).”

*Reviewer #2:*

In this manuscript, Matsumoto et al. investigated the response of midbrain dopaminergic neurons to an aversive stimulus in a variety of contexts. Although dopamine is thought to be important for learning from both reward and punishment, there has been disagreement about how dopaminergic neurons integrate aversive and rewarding stimuli. The authors show that dopaminergic activity is inhibited by an air puff and an air puff-predicting odor in low reward contexts. The air puff response is modulated by expectation and individual dopaminergic neurons are able to integrate the value of reward and punishment. Finally, they show that manipulating reward context affects dopaminergic responses to punishment, such that in high reward contexts there is actually an excitatory response to punishments. The latter condition appears to replicate various previous studies. The data provides a valuable contribution to the debate over how dopaminergic neurons respond to aversive stimuli. The use of optogenetic tagging allows for compelling identification of dopaminergic neurons and the authors tested several task structures in order to reconcile discrepancies in the field.

*There are several ways in which the analysis and presentation should be improved:*

*1) In Figure 2 the authors conclude that in a high reward context, dopaminergic neurons respond to aversive stimuli with short-latency excitation "masked inhibitory responses" such that there is no longer a significant difference between responses to odor B and odor C. The importance of reward context is a major conclusion of the paper but its complete disruption of aversive signaling is confusing (and not entirely convincing) as presented. Why not analyze the different temporal phases of the response separately? It appears that if the responses were compared during the late inhibitory phase of the response (evident in Figure 2), then there would indeed be a difference in the response to odor B and odor C in both the high reward and low reward contexts. In other words, the simplest explanation of the effect of "high reward context" is to create a short-latency excitatory response to the aversive stimulus. The authors argue this response is different from the short-latency 'generalization' response described by Schultz and colleagues, but not clear what that means. Specifically, in the Discussion it is noted that a high reward context "compromised the monotonic value coding of dopamine neurons," but this was not clearly explained or analyzed. Additionally, the argument that the excitatory response to punishment in a high reward context is not due to generalization is unclear. The authors write that "dopamine neurons did not simply add the same amount of excitation on top of the monotonic value coding." In fact, in Figure 2, it seems like there is an excitatory "generalization" response that is being added to the pause in firing to the CS-.*

Thank you for the criticism and suggestions. We now added these analyses in Figure 4—figure supplement 2. Please also see our responses to the editor’s point #1 and reviewer 1’s point #2 and #3.

*2) Although there is a clear increase in the amplitude of the response to air puff when it is unpredicted, there is large variability in the response to an omitted air puff. Although on average there is a very small increase in firing rate, a very large number of neurons decrease their activity in response to air puff omission (Figure 3). The authors should discuss the heterogeneity of this response, which appears to complicate their conclusions.*

We agree that air puff omission responses are very small and variable. Because we only mentioned the smallness, we now mentioned the variability in Results.

“To further examine whether dopamine neurons showed prediction error coding for aversive events, we compared the firing rate during the outcome period in air puff omission trials with that in trials that predict nothing. We found that the omission of a predicted air puff slightly but significantly increased firing rates, compared to no change in nothing trials (Figure 2) although we observed variability in air puff omission responses.”

The air puff omission responses are correlated with air puff prediction and we did not observe clear clusters using these factors. We now added this analysis in Figure 3—figure supplement 1.

*3) It seems surprising that there is no correlation between the response to unpredicted air puff and unpredicted water delivery (Figure 5). In Eshel et al. (2016) there is a correlation between predicted air puff and unpredicted water responses as well as between the air puff cue and the unpredicted water responses. So why wouldn't there be a correlation between unpredicted air puff and reward? Doesn't this mean that predicted and unpredicted air puff responses are themselves uncorrelated, which would contradict the earlier panels in Figure 5?*

As we found in the present study, the negative value of aversive stimulus is well represented in low reward contexts and the responses to aversiveness-related events are correlated with one another in these conditions. Importantly, the previous study (Eshel et al., 2016, Nat Neurosci) was conducted using a high reward context. A previous study found that the inhibition phase after excitation in biphasic responses is not related to aversiveness but to “no reward” (Fiorillo, 2013, Science). Consistent with this idea, we observe that in high reward contexts, cues predicting nothing often caused inhibition in dopamine neurons instead of no responses. In the previous study (Eshel et al., 2016, Nat Neurosci), the authors used cues which signaled either water or air puff (no cue signaling nothing). In this sense, the cue signaling air puff actually signals “no reward”. We interpret that the correlation which authors observed in responses between unpredicted water and air puff CS was mainly due to this “no reward” responses. Further, as the paper stated, the correlation was weak and not statistically significant.

The correlation between predicted air puff and unpredicted water in the previous study (Eshel et al., 2016, Nat Neurosci) was also complicated due to similar reasons. We examined author’s data more carefully and found that the correlation came from rebound after inhibition, but not inhibition itself; as Fiorillo reported (Fiorillo et al., 2013a, J Neurosci; Fiorillo et al., 2013b, J Neurosci), dopamine neurons actually often showed three phases (excitation, inhibition and excitation). We observed high correlation between this rebound and water responses but not between inhibition and water responses in high reward contexts.

Our present study is improved to more directly test the relation between rewarding and aversive value representations in dopamine neurons. We observed a correlation both within aversiveness-related representation and within reward related-representation but not between these. We now added this statement more clearly in Results (subsection “Homogeneous response function of dopamine neurons”, third paragraph) and Discussion (subsection “Integration of aversiveness and reward in dopamine neurons”, last paragraph).

*4) The protocol for the classical conditioning experiment conducted for Figure 2—figure supplement 1 is unclear. It appears that the only difference from the high reward task is the probability of punishment following odor C which is 80% instead of 90%, yet the excitatory component of the response to the punishment-predicting CS appears considerably larger than in Figure 2.*

Thank you for pointing this out. Yes, this is a very similar task except that the reward probability was slightly higher and the training procedure was slightly different. In this task (high reward probability task 2), we used 6.5% unpredicted reward delivery in comparison with 6% unpredicted water delivery in the high reward probability task. In addition, in the task 2, the odor predicting 90% water during experiments was same with the odor predicting 100% water during the initial training (a few days). It might be possible that animals might predict water with slightly higher probability than 90% during recording in the task 2 (i.e. higher reward context in the task 2). We now added a detailed task description in Materials and methods (subsection “Behavioral task”, second and last paragraphs). We did not see a significant difference in air puff CS responses between these two tasks (p = 0.19, unpaired *t*-test, n = 17 versus 17). We agree that this was confusing and we now added the pooled data for responses to air puff- and nothing-predicting CSs in Figure 4—figure supplement 1.

*5) Not clear exactly what the eye blink response analysis achieves in the paper, nor why eye blinks were analyzed but not anticipatory licking.*

Please see our responses to the editor’s point #2 and reviewer 1’s point #1. The main message from the analysis on the neural activity-behavior correlations is to point out the importance of not only the contexts but also behavior in order to understand potential causes of the discrepancies between studies. For reward responses, there are not much physiological discrepancies, which may be due to strong rewarding effects of water, compared to relatively mild aversion with mild air puff. Although anticipatory licking is much more consistent, we observed similar tendency of dopamine responses in relation to anticipatory licking. We now added these results in Figure 5 and Figure 5—figure supplement 4 and in Results (last paragraph).

*6) There is no reported method for the correction for multiple comparisons in multiple instances when more than 2 groups are being analyzed in the same panel. Additionally, a two-way ANOVA should be used to analyze the effect of reward probability and anticipatory blinking in Figure 4.*

Thank you for the suggestions. A two-way ANOVA cannot be used in Figure 5 (the original Figure 4) because these are two different data sets. We agree that the figure was confusing and we modified the graphs (boxplots derived from two different tasks were displaced separately). We applied Bonferroni’s correction for significance tests with multiple comparisons and added this statement in Materials and methods.

*Reviewer #3:*

*In this manuscript Matsumoto and colleagues present a series of experiments that address the issue of how dopamine neurons respond to aversive events. They make a number of important and interesting observations. On the whole I thought the manuscript was well-written. I have a few concerns that I list below:*

*1) I feel that the Introduction would benefit from some elaboration in a few places.*

Thank you for the advice. We expanded some discussions in Introduction.

*Introduction, fourth paragraph: should make it clear that Oleson et al. (2012) measured dopamine release using FSCV.*

RESPONSE: We added this information (Introduction, fourth paragraph).

*Introduction, fifth paragraph: I am not clear how Roitman et al. (2008) resolved the issue and this was published several years earlier, so the wording is confusing in my view.*

We rephrased the sentence (Introduction, fifth paragraph).

*Introduction, fifth paragraph: should make it clear that Lerner et al. (2015) measured calcium activity, not action potential firing.*

We added this information (Introduction, fifth paragraph).

*Introduction, last paragraph: this paragraph alludes to the issue of neurochemical identity in a way that I think should be more explicit since it is of particular relevance to this study. For example, Ungless et al. (2004) showed that some VTA neurons that are excited by aversive events are in fact not dopaminergic. It is noted, for example by Schultz (2016) in a recent review, that Matsumoto & Hikosaka (2009) recorded many neurons in areas where dopamine neurons were rare. This more strongly justifies the use of the opto-tagging approach that the authors have taken.*

Thank you for letting us know! We now added these sentences in Introduction (eighth paragraph) and emphasized more about identification in Abstract and Discussion.

However, regarding the citation of Schultz’s statement on potential problems in recording locations in some previous studies, our opinion is that we are not in a position to strongly support either view (Schultz’s or the original researchers’). We, therefore, would like to not include this sentence (“Furthermore, Schultz has argued that some previous recording studies may not have targeted areas rich in dopamine neurons (Schultz, 2016)”) if the reviewers are fine with it.

*2) One general concern I had was the low numbers of mice in several of the experimental groups (e.g., Figure 1=3).*

Please see the answer to the editor’s point #3. The number of animals is comparable to other electrophysiology studies of dopamine neurons using monkeys (Fiorillo, 2013, Science; Fiorillo et al., 2013a, J Neurosci; Fiorillo et al., 2013b, J Neurosci; Kobayashi and Schultz, 2014, Curr Biol; Matsumoto and Hikosaka, 2009, Nature) and mice (Cohen et al., 2012, Nature; Eshel et al., 2015, Nature; Tian and Uchida, 2015, Neuron). We also observed similar tendencies by using data from two instead of three animals (after leaving out one out of three animals). We added this data in Figure 1—figure supplement 1.

*3) How do the authors know that the mice can tell the difference between the low and high reward situations?*

This is an important point. We observed that the level of anticipatory licking to reward cue was different between these two contexts (stronger with the reward CS predicting high probability of water). We have added this result in Figure 5—figure supplement 4B. Dopamine responses are also different between these two contexts. However, how reward contexts changed mouse conditions (consciously or unconsciously) is an important question to be resolved in future experiments.

*4) In the subsection “Integration of aversiveness and reward in dopamine neurons”, first paragraph: add the caveat that activation of meso-cortical dopamine neurons and dorsal raphe dopamine neurons can induce conditioned-place aversion (Lammel et al. 2014 & Matthews et al. 2016).*

Thank you for pointing out these references. We have added these references in Discussion(subsection “Integration of aversiveness and reward in dopamine neurons”, first paragraph).

*4) In the subsection “Diversity of dopamine neurons”: note also that Brischoux et al. (2009) reported that ventromedial dopamine neurons exhibited aversive activations.*

Thank you. We have added this information in Discussion (subsection “Diversity of dopamine neurons”, first paragraph).

*5) I found some of the red/purple colour schemes difficult to tell apart. I would suggest some different colour schemes (particularly for Figure 1).*

Thank you for pointing this out. We changed the blue and purple colors lighter in Figures.

[Editors' note: further revisions were requested prior to acceptance, as described below.]

*The manuscript has been improved but there are some remaining issues regarding its readability that need to be addressed before acceptance, as outlined below by Reviewer 2. After consultation between the reviewers, both thought that increased readability and explanation of why each experiment was performed would greatly enhance the clarity and impact of the study.*

Thank you for the advice. We added one paragraph specifically devoted to explanations of each experiment and comparisons between task conditions at the beginning of the Results, together with Figure 1 and Table 1.

*Reviewer #2:*

*I felt that the reviewers were responsive to our comments and that this study provides a valuable contribution to the discussion regarding the integration of rewarding and aversive cues in dopamine neurons.*

*My only remaining comment is that I find the text in the Results and Discussion a bit difficult to follow. There are a number of experiments/analyses, and as it currently stands, the motivation for each experiment/analysis, as well as the relevant conclusions, is not always completely transparent. Therefore, the authors could consider editing the Results/Discussion for improved readability.*

We added explanations of each experiment as mentioned above. We also rearranged Discussion and deleted some sentences for better readability.